

**Contrasting Inland-Coastal Aerosol Mixing States: An Entropy-Based**
**Metric for CCN Activity**
Jingye Ren[1,2], Wei Xu[3], Ru-Jin Huang[1*], Fang Zhang[4*], Ying Wang[1], Lu Chen[5], Jurgita
Ovadnevaite[6], Darius Ceburnis[6], Colin O'Dowd[6]
*[1]State Key Laboratory of Loess Science, Institute of Earth Environment, Chinese*
*Academy of Sciences, Xi'an, 710061, China,*
*[2]Xi'an Institute for Innovative Earth Environment Research, Xi'an, 710061, China,*
*[3]State Key Laboratory of Advanced Environmental Technology, Institute of Urban*
*Environment, Chinese Academy of Sciences, Xiamen, 361021, China,*
*[4]School of Civil and Environmental Engineering, Harbin Institute of Technology,*
*Shenzhen, 518005, China,*
*[5]School of Ocean and Geographic Science, Yancheng Teachers University, Yancheng*
*224051, China,*
*[6]School of Natural Sciences, Centre for Climate & Air Pollution Studies, Ryan Institute,*
*University of Galway, University Road, Galway, Ireland*
Corresponding author: Ru-Jin Huang, *rujin.huang@ieecas.cn;* Fang Zhang,
*zhangfang2021@hit.edu.cn*





**Abstract**

Simplified assumptions of aerosol hygroscopic mixing states in modeling studies

often introduce substantial uncertainties in estimating cloud condensation nuclei (CCN)
concentrations and their climatic impacts. This study systematically investigates the
contrasting relationships between mixing states and CCN activity by comparing
ambient measurements from inland and coastal sites. We show distinct seasonal
variations of the particles mixing state. In winter, externally mixed particles dominated
both sites, with comparable mixing state indices ($\chi$) of 0.38±0.12 and 0.39±0.09
respectively for coastal air masses and inland air. However, summer measurements
showed pronounced differences: photochemical processes promoted significantly
higher internal mixing in coastal aerosols ($\chi$=0.69±0.19), whereas inland $\chi$ values only
increased moderately to 0.47±0.12. A universal logarithmic correlation was identified
between the critical diameter ($D_{cri}$) characterizing CCN activity and $\chi$ ($D_{cri}$ = -
32.15ln($\chi$)+84.71, Pearson r = -0.74), but with distinct decrement rates for coastal vs.
inland aerosols. Our further quantitative analysis reveals a 0.1 increase in $\chi$ enhanced
winter CCN concentrations ($N_{CCN}$) by 39–65% under typical cloud supersaturations,
whereas this effect diminished to ~9% in summer. These results underscore that mixing
states exert more pronounced control over $N_{CCN}$ in diverse environments. Our work
provides critical constraints for parameterizing fine aerosols CCN activity in climate
models, thereby reducing uncertainties in aerosol–climate effect estimations.



## 1. Introduction


Atmospheric cloud condensation nuclei (CCNs) are complex mixtures of organic
and inorganic components. Their chemical and physical properties make quantifying
aerosol-cloud interactions challenging (Liu et al., 2018; Rosenfeld et al., 2019; Xu et
al., 2022, 2024; Virtanen et al., 2025), introducing uncertainties into climate effect
assessments (Charlson et al., 1992; Shrivastava et al., 2017; IPCC, 2021; Manavi et al.,
2025; Chen et al., 2022). Accurate climate model predictions of aerosol impacts require
understanding aerosol mixing states under different atmospheric conditions and their
effects on CCN activity (Ching et al., 2016; Zheng et al., 2021). Current models often
oversimplify mixing states by assuming pure internal or external mixing (Winkler, 1973;
Zheng et al., 2021; Stevens et al., 2019; Riemer et al., 2019). This is problematic
because mixing states directly determine particle hygroscopicity and CCN estimates
(Wang et al., 2010; Ren et al., 2018). For example, CCN activity for internal-mixed
aerosols rely more on inorganic components, while external mixtures are more sensitive
to organic matter (Ren et al., 2018; Bhattu et al., 2015). Such simplifications can lead
to significant errors, e.g., Sotiropoulou et al. (2007) found that mixing state assumptions
caused two-fold $N_{CCN}$ estimation errors in global models.
Systematic observations across diverse environments are critical because aerosol
mixing states exhibit pronounced spatial-temporal variations (Ye et al., 2018; Liu et al.,
2025; Hughes et al., 2018). For example, continental and coastal regions present
contrasting scenarios (Ramachandran et al., 2016). The continental areas are dominated
by anthropogenic emissions, where aerosol aging is driven by industrial and traffic-



related pollutants (Huang et al., 2014; Ren et al., 2023). Particles here undergo
progressive internal mixing via photochemical reactions and coagulation, altering their
hygroscopic properties (Ervens et al., 2010). While the coastal regions feature dynamic
interactions between marine aerosols (e.g., sea salt) and continental pollutants (Schill
et al., 2015; Collins et al., 2013; Cheung et al., 2020). Seasonal shifts in air mass sources
(e.g., marine vs. continental dominance) create unique mixing state patterns (Xu et al.,
2020, 2021a). For instance, summer photochemical processes in coastal areas can
enhance internal mixing, while winter often retains more external mixing due to stable
atmospheric conditions.
However, the aerosols in continental and coastal regions have distinct climate
feedback mechanisms (Bellouin et al., 2019; Pan et al., 2022; Gong et al., 2023). The
continental aerosols influence regional cloud formation, while coastal aerosols affect
marine boundary layer clouds that are key components of global climate systems (Liu
et al., 2018). But the current models lack regional-specific mixing state parameters and
usually assume uniform mixing in both environments. This could lead to large
uncertainties in predicting CCN concentrations, highlighting the need for site-specific
observations.
Recent studies have used the mixing state index ($\chi$) to characterize aerosol
heterogeneity (Zheng et al., 2021; Ching et al., 2017; Yuan et al., 2023), but cross-
environment comparisons remain limited. By integrating inland and coastal
measurements, this study will focus on addressing two key gaps, (1) How continental
vs. marine-dominated environments shape aerosol mixing states and CCN activity; (2)





Whether χ-based CCN parameterizations show regional dependencies, providing
critical constraints for climate models.
**2. Data and Methods**
**2.1 Field Campaigns**
The inland atmospheric measurements were conducted for two periods from 16
November to 6 December and 29 May to 13 June, respectively in urban Beijing, at the
Institute of Atmospheric Physics, Chinese Academy of Sciences (IAP, 39.97° N,
116.37° E). This urban site exhibited highly variable aerosol populations dominated by
local anthropogenic sources including vehicular, cooking emissions, and residential
heating. Coastal measurements were performed at the Mace Head atmospheric research
station (MHD, 53.33° N, 9.90° W) from 1 November 2009 to 30 January 2010 and 11
to 31 August 2010, which located on the west coast of Ireland. Aerosol particles here
experience alternating influences from polluted continental and clean marine
atmospheres. The map of the sites was shown in Figure 1. More details about the
campaigns were given in Fan et al. (2020) and Xu et al. (2021a).

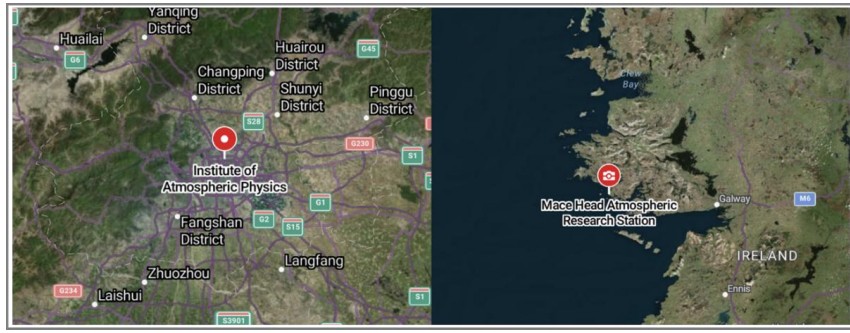


**Fig 1.** Map of the sites in the Inland of the Institute of Atmospheric Physics (IAP) and



Coastal of Mace Head (MHD). (© Google Maps, https://maps.google.com/, last access:
2 April 2025).

## 2.2 Instrumentation

### Hygroscopicity measurements

The particle hygroscopicity at both sites was characterized using the humidified
tandem differential mobility analyzer (HTDMA). The hygroscopic growth factor (Gf),
defined as the ratio of the particle diameter at the fixed RH (90%) and dry diameter set
in this study for 40, 80, 110, 150, 200 nm at IAP and 35, 50, 75, 110 and 165 nm at
MHD, respectively. The Gf probability density function (Gf-PDF) was derived using
the TDMAinv algorithm (Gysel et al., 2009).
Here for each particle size, the hygroscopicity parameter $\kappa$ can be subsequently
calculated using $\kappa$-Köhler theory (Petters and Kreidenweis, 2007):
$$\kappa = (Gf^3 - 1) \cdot \left[\frac{1}{RH} \exp\left(\frac{4\sigma_s M_w}{RT\rho_w D_d Gf}\right) - 1\right] \qquad (1)$$
where RH is the HTDMA relative humidity (90% set in the instrument), $\sigma_{s/a}$ is the
surface tension of pure water (0.072 mN m$^{-1}$), $M_w$ and $\rho_w$ are the molecular weight and
the density of pure water, $R$ is the gas constant, and $T$ is the absolute temperature, $D_d$ is
the droplet diameter.
Then, the $\kappa$-PDF is obtained and normalized as $\int_0^\infty c(\kappa)d\kappa = 1$, where $c(\kappa)$ is
normalized as $\kappa$-PDF. Further it was used to calculate the particle population
heterogeneity (Calculation seen in Section 2.3).

### Chemical components



For the inland atmospheric measurements, the non-refractory submicron aerosol
(smaller than 1μm, NR-PM$_1$) chemical composition was quantitatively characterized
using the Aerodyne High-Resolution Time-of-Flight Aerosol Mass Spectrometer (HR-
ToF-AMS) (DeCarlo et al., 2006), including sulfate (SO$_4^{2-}$), nitrate (NO$_3^-$), ammonium
(NH$_4^+$), chloride (ChL) and organics (Org). The black carbon (BC) mass concentration
was determined from the light absorption with a seven-wavelength aethalometer (AE33,
Magee Scientific Corp.).
Measurements of PM$_1$ in the coastal atmosphere were also performed by the HR-
ToF-AMS, including major inorganic salts (non-sea-salt sulfate, nss-SO$_4^{2-}$;
methanesulfonic acid, MSA; NO$_3^-$; NH$_4^+$) and organic matter. The instrument operation
and calibration have been described in previous studies (Ovadnevaite et al., 2014; Xu
et al., 2019).
**Aerosol number size distribution and CCN number concentration**
Particle number size distributions (PNSD) were measured using an integrated
system consisting of a Differential Mobility Analyzer (DMA; model 3081, TSI Inc.)
coupled with a Condensation Particle Counter (CPC; model 3772, TSI Inc.). During the
measurements at IAP, the PNSD covered the size range of 10-550 nm with a 5-minute
time resolution. It scanned size range of 20-500 nm at MHD with a 10-minute temporal
resolution. The CCN number concentrations were quantified at both sites using a
Droplet Measurement Technologies CCN counter (DMT-CCNc) (Lance et al., 2006).
The instrument's supersaturation (SS) settings were carefully calibrated before and after
each campaign using ammonium sulfate aerosol following Rose et al. (2008).



### 2.3 Calculation the heterogeneity for aerosol particles


To characterize the heterogeneous distribution of the hygroscopic and non-
hygroscopic components in populations (Chen et al., 2022), we calculated the mixing
state index ($\chi$) using the $\kappa$-PDF, following the methodology of Yuan et al. (2023). Two
surrogate groups in a population of $N$ aerosol particles were assumed (Zheng et al.,
2021). One surrogate group consists the non-hygroscopic species with $\kappa_{NH}$ of 0.01
and another group contains the hygroscopic species with $\kappa_H$ of 0.6 (Yuan et al., 2023;
Ching et al., 2017). At the coastal MHD site, we accounted for the enhanced
hydrophilicity of marine aerosols by additionally testing $\kappa_H$ values of 0.7 and 0.8 (Fig.
S1). While these variations in $\kappa_H$ introduced a mean uncertainty of 8% in $\chi$ values, it
did not significantly affect the seasonal or site comparisons. The volume fraction of two
surrogate groups can be calculated based on the total $\kappa$ according to the Zdanovskii–
Stokes–Robinson (ZSR) mixing rule (Zdanovskii, 1948; Stokes et al., 1966).
The mixing state index $\chi$ is defined as the affine ratio of the average particle species
diversity ($D\alpha$) and population species diversity ($D\gamma$) as:

$$\chi = \frac{D_\alpha - 1}{D_\gamma - 1} \tag{2}$$

The average per-particle species diversity $D\alpha$ can be calculated as follows. First,
the mixing entropies at bin $i$ ($H_i$) are determined according to equation (3),

$$H_i = -P_{i,NH} \times ln P_{i,NH} - P_{i,H} \times ln P_{i,H} \tag{3}$$

where $P_{i,NH}$ and $P_{i,H}$ are the volume fraction of each group for the $\kappa$-PDF with X bins
at bin $i$ ($i=1,2,…X$), and can be determined from the $P_{i,NH} + P_{i,H} = 1$ and
$P_{i,NH} \times \kappa_{NH} + P_{i,H} \times \kappa_H = \kappa_i$. Here $\kappa_{NH} = 0.01, \kappa_H = 0.6$; $\kappa_i$ represents the



hygroscopicity parameter at bin $i$.
Based on the assumption that particles in the same diameter have the same mixing
entropy $H_\alpha = \sum_{j=1}^{N} P_j \times H_j$, $P_j = \frac{V_j}{V_{total}} = \frac{1}{N}$; the per-particle mixing entropies $H_\alpha$ is
determined according to equation (4),

$$H_\alpha = \sum_{i=1}^{X} H_i \times c(\kappa)_i \times \Delta\kappa \qquad (4)$$

where $c(\kappa)_i$ is the probability density of the normalized $\kappa$-PDF at bin $i$, and $\Delta\kappa$
represents the bin width. Then, the average per-particle species diversity $D\alpha$ can be
determined as $D_\alpha = e^{H_\alpha}$;
The bulk population species diversity $D\gamma$ can be calculated as follows. First, the
aerosol population of the mixing entropy can be calculated as equation (5):

$$H_\gamma = -P_{NH} \times lnP_{NH} - P_H \times lnP_H \qquad (5)$$

where $P_{NH}$ and $P_H$ are the volume fraction of the non-hygroscopic and hygroscopic
components in the population, and can be calculated by equation (6) and (7):

$$P_{NH} = \sum_{i=1}^{X} P_{i,NH} \times c(\kappa)_i \times \Delta\kappa \qquad (6)$$

$$P_H = \sum_{i=1}^{X} P_{i,H} \times c(\kappa)_i \times \Delta\kappa \qquad (7)$$

Then, the bulk population species diversity $D\gamma$ can be determined as $D_\gamma = e^{H_\gamma}$.
Here, the definition of surrogate species as supersets encompassing hygroscopicity
heterogeneity implies that the heterogeneity parameter $\chi$ ranges from 0 to 1. When the
mixing index $\chi$ approaches 0, it indicates a completely segregated state where
hygroscopic and non-hygroscopic species reside in distinct particles. While for the case
the mixing index $\chi$ to be 1 represents that the non-hygroscopic and hygroscopic species
distributing homogeneously throughout the aerosol population.



## 3. Result and Discussion

### 3.1 Comparison of the heterogeneity in the inland and coastal atmosphere

To characterize the hygroscopic heterogeneity of atmospheric aerosols, Figure 2 depicts variations in mixing state metrics ($D\alpha$, $D\gamma$, $\chi$) and the hygroscopic parameter ($\kappa_{gf}$) across particle size distributions. For inland aerosols, $D\alpha$ and $\chi$ decrease with increasing particle diameter, accompanied by higher $\kappa_{gf}$ values. This trend indicates that inland particle populations tend to homogenize into hygroscopic compositions through primary particle aging or secondary formation processes (Liu et al., 2025; Chen et al., 2022; Zhong et al., 2022). In contrast, coastal particles exhibit a non-monotonic pattern: $D\alpha$ and $\chi$ decrease for Aitken-mode particles (<100 nm) but increase for accumulation-mode particles. The $\kappa_{gf}$ shows consistent size-dependent increases in both winter and summer campaigns.

Notably, the mixing state metrics exhibit a pronounced minimum at 75 nm particles, influenced by distinct mechanisms: winter minima reflect the high sea salt fraction, while summer minima are driven by anthropogenic organic matter (Cheung et al., 2020; Xu et al., 2021a). Lower winter $\chi$ values—coupled with broader $\kappa$-PDF distributions—indicate stronger external mixing and compositional diversity compared to summer (Fig. S2). Seasonal $\chi$ and $\kappa_{gf}$ disparities are more pronounced at the coastal site, primarily driven by the seasonal alternation of marine and anthropogenic emission sources.





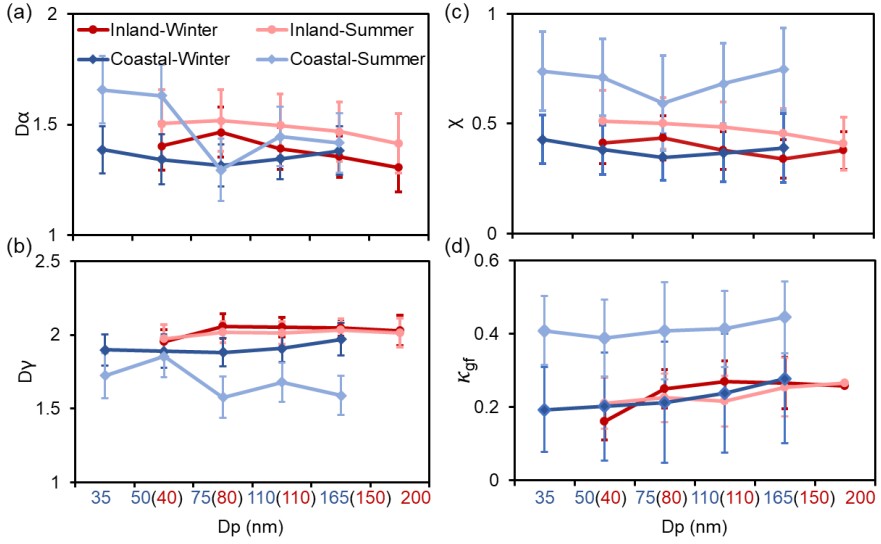

**Fig 2.** Mean values of the Dα (a), Dγ (b), χ (c) and $\kappa_{gf}$ (d) for aerosols of five diameters during winter and summer periods in Inland (IAP) and Coastal (Mace Head) sites.

Ultrafine particles (40 nm inland vs. 35 nm coastal, Aitken mode) and larger particles (150 nm inland vs. 165 nm coastal, accumulation mode) were selected to investigate distinct evolutionary processes of aerosol heterogeneity (Fig. 3 and Fig. S3). With the increasing of PM concentration during winter, the variation in χ values exhibit only minor both at the inland and coastal sites, generally fluctuating between approximately −0.04 and 0.08 (Fig. 3a and b). Inland accumulation-mode particles show a modest increase in χ, corresponding with a higher proportion of inorganic salts. Conversely, at coastal sites, the composition fraction shifts from a sea-salt dominance toward organic matter, accompanied by a ~20 % increase in nitrate content (Fig. 3b). In summer, the variation of χ with PM concentration becomes markedly pronounced at both inland and coastal stations. For example, χ for 40 nm particles decreases as PM



increases at inland sites (Fig. 3c). The elevated particle heterogeneity mainly arises
from the locally primary emissions and photochemically driven new particle formation.
In contrast, χ for 150 nm particles increases from ~0.40 to ~0.60 with rising PM,
reflecting enhanced secondary formation and internal mixing during pollution process
that render the particle population more homogeneous. At coastal sites, χ declines with
rising PM by approximately 0.37 for 35 nm particles and 0.24 for 165 nm particles,
mirroring the shift in chemical composition makeup from inorganic dominance to
greater organic content (Fig. 3d).

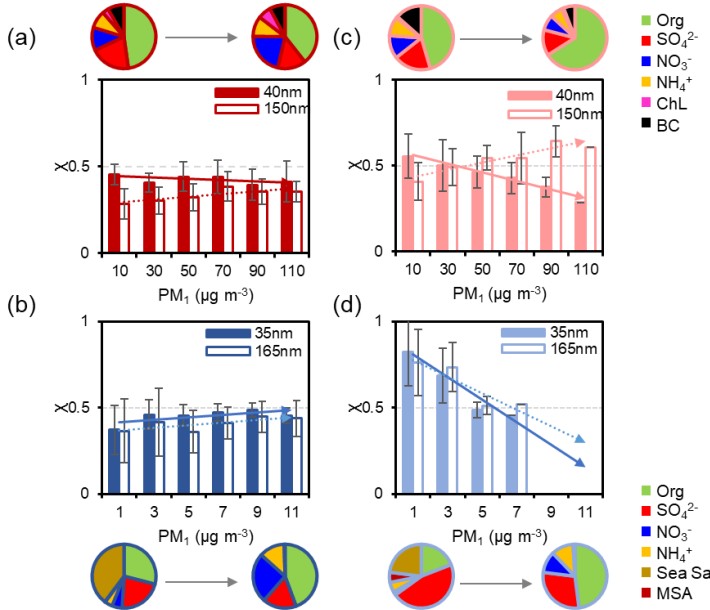


**Fig 3.** Variation of the average χ of 40 nm and 150 nm in inland and 35 nm and 165 nm

in coastal site with the particle mass concentration in Inland-winter (a), Inland-summer
(b), Coastal-winter (c) and Coastal-summer (d). The pie charts represent the average
mass fraction during four field campaigns.



Figure 4 illustrates pronounced diurnal variations in mixing state metrics (Dα, Dγ,
Gf-PDF, χ) between inland and coastal atmospheres. In the inland atmosphere, winter
exhibited steeper declines in Dα and χ during evening rush hours than summer,
indicating a higher fraction of non-hygroscopic particles (40 nm) from fresh traffic
emissions (Fig. 4a1). Concurrently, reduced Dγ values suggest that the bulk population
consists of uniformly distributed less-hygroscopic (LH) components (Fig. 4c1). Aitken
mode particles showed bimodal and broader Gf-PDF distributions, corresponding to
cooking activities (11:00–13:00 LT) and traffic peaks (17:00–20:00 LT) (Cai et al.,
2020). Midday photochemical aging promoted more internally mixed aerosols (Yang et
al., 2012; Liu et al., 2025), as evidenced by increasing Dα at the urban site (Fig. 4b1).
Conversely, accumulation-mode particles showed minimal diurnal variations,
suggesting stable relative proportions of LH and more-hygroscopic (MH) components
in inland aerosols across seasons.

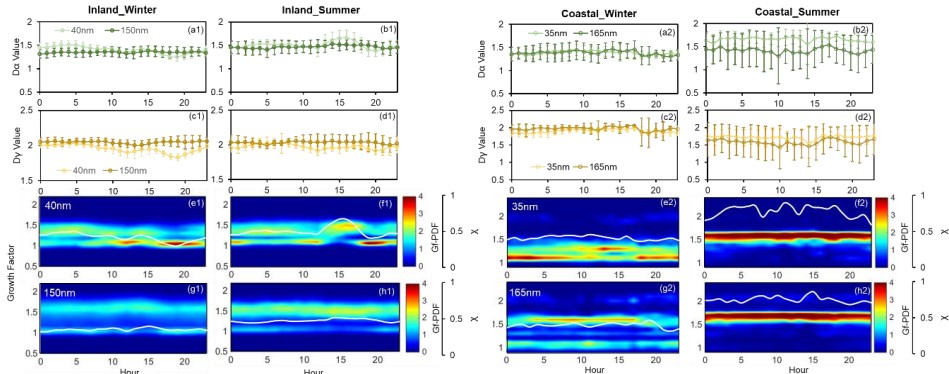

**Fig 4.** The variation of Dα, Dγ, Gf-PDF, and χ during winter and summer periods for
40 nm and 150 nm aerosols in Inland (a1-h1) and for 35 nm and 165 nm aerosols
Coastal site (a2-h2).





For the coastal atmosphere, the mixing state metrics ($D\alpha$, $D\gamma$, and $\chi$) of Aitken and
accumulation mode particles in winter exhibited analogous diurnal patterns,
characterized by a descending trend at nightfall. This corresponds to an enhanced modal
distribution of near-hydrophobic (NH) particles at 35 nm and more-hygroscopic (MH)
particles at 165 nm. In summer, $D\alpha$ and $D\gamma$ both trended downward during daytime,
with the decline of $D\gamma$ being more pronounced. A conspicuous seasonal discrepancy
between Aitken and accumulation mode particles was observed in this region (Fig. 4a2–
h2), where the mixing state index $\chi$ increased incrementally from winter to summer.
Specifically, the mean $\chi$ for 35 nm particles escalated from 0.42 to 0.80, and for 165
nm particles, it rose from 0.39 to 0.76. This trend demonstrates a strong alignment with
the spread factor documented by Xu et al. (2021a, b).
The Gf-PDF diurnal profiles of Aitken mode particles displayed a bimodal and
broadened distribution, corresponding to a less-hygroscopic (LH) mode of biogenic
origin during nighttime and a more-hygroscopic (MH) mode dominated by sea salt
(comprising 55% number fraction) during daytime. Analogously, accumulation mode
particles exhibited bimodal distributions with a higher proportion of MH mode during
daytime, primarily attributed to the prevalence of sea salt and non-sea-salt sulfate (nss-
sulfate) in the coastal atmosphere (Xu et al., 2020). In contrast, summer observations
revealed that Gf-PDFs of both Aitken and accumulation mode particles transitioned to
unimodal distributions, signifying more homogeneous mixing of LH and MH
components within individual particles. This uniformity is linked to processes including
sulfuric acid condensation, admixture of sulfate with biogenic organic matter (Xu et al.,





2021a), as well as photochemical oxidation and atmospheric aging (Jimenez et al.,

2009).

## 3.2 Dependence of the aerosol properties on the mixing state

The mixing state of particle populations undergoes dynamic transformations
during atmospheric aging, profoundly influencing their CCN activity. Unlike prior
studies that assumed mixing states based on chemical component fractions (Yang et al.,
2012; Padró et al., 2012; Ren et al., 2018), this work employs the entropy-derived
mixing state index $\chi$, which quantifies the distribution of hygroscopic and non-
hygroscopic species (Zheng et al., 2021; Ching et al., 2017). We systematically
investigate how aerosol properties evolve with changing $\chi$. Figure 5 illustrates the
dependency of aerosol characteristics on $\chi$ (ranging from 0 to 1 in 0.1 increments),
presenting key insights into particle size and chemical composition—two fundamental
determinants of CCN activity (Ren et al., 2018).
As $\chi$ increases, the peak diameter ($D_{peak}$) of the particle number size distribution
(PNSD) shifts toward larger sizes (Fig. 5a and Fig. S4), while peak concentrations occur
within the intermediate $\chi$ range (0.3–0.6). This trend indicates that CN number
concentration ($N_{CN}$) first increases, driven by primary emissions and new particle
formation, then decreases due to mixing and aging processes (Fig. 5b). Notably, inland
summer $N_{CN}$ exhibits a sustained slight increase, linked to frequent new particle
formation events and subsequent particle growth.
The critical diameter ($D_{cri}$)—defined as the minimum size for activation at a given



supersaturation—depends on the mass fraction of soluble components (Petters and
Kreidenweis, 2007). Using a typical cloud supersaturation of 0.2% as a case study, Fig.
5c shows that $D_{\mathrm{cri}}$ decreases with increasing soluble species (e.g., sulfate, nitrate) in the
inland atmosphere. In contrast, coastal $D_{\mathrm{cri}}$ exhibits nonlinear variations with χ: high
external mixing (low χ) elevates $D_{\mathrm{cri}}$ due to dominant organic components, reducing sea
salt particle fractions. As χ increases, the mass fraction of non-sea-salt sulfate (nss-
sulfate) rises, enhancing activation potential by decreasing $D_{\mathrm{cri}}$.

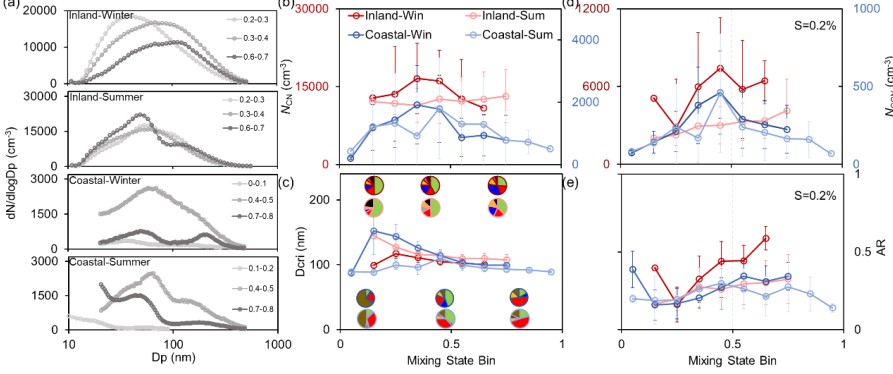


**Fig 5.** Variation of the average particle number size distribution (PNSD) with the
mixing state index χ (a), variation of the $N_{\mathrm{CN}}$ with the χ (b), variation of the $D_{\mathrm{cri}}$ and
mass fraction of chemical composition with the χ (c), variation of the $N_{\mathrm{CCN}}$ and
activation ratio (AR) at $S$=0.2% with the χ (d-e).
The dependence of CCN activity at 0.2% supersaturation on mixing state index χ
reveals distinct inter-atmospheric differences, as shown in Fig. 5d-e. In the inland
atmosphere, $N_{\mathrm{CCN}}$ at $S$=0.2% demonstrates a monotonic increasing trend with χ,
attributed to the synergistic effects of rising $N_{\mathrm{CN}}$ and decreasing $D_{\mathrm{cri}}$ (Fig. S5). By
contrast, coastal $N_{\mathrm{CCN}}$ follows a pattern analogous to $N_{\mathrm{CN}}$, with peak concentrations





shifting toward higher χ values. This highlights the dominant role of particle size effects
in enhancing CCN concentrations under marine-influenced conditions (Perkins et al.,

2022).

Two distinct $D_{cri}$-χ trends underpin these disparities: one remains stable, driven by

the inherent hygroscopicity of sea salt, while the other exhibits steep $D_{cri}$ declines
associated with anthropogenic pollution as internal mixing intensifies. These
discrepancies are further manifested in the nonlinear $D_{cri}$-χ relationship. The activation
ratio (AR)—quantifying aerosol cloud droplet formation potential at fixed
supersaturation—also varies by site (Fig. 5e). Notably, inland winter AR shows a
marked increase with χ, likely due to enhanced $N_{CCN}$ from the elevated inorganic
fraction under higher mixing states (Fig. 3). Conversely, the inorganic fraction
decreases during other sampling periods, dampening AR growth.

### 3.3 Impact of the mixing state on the CCN activity

To better interpret the impact of mixing state on CCN concentrations, Fig. 6

quantifies the relative change in $N_{CCN}$ at S=0.2% as mixing state index χ increases,
contextualizing how CN concentration and chemical compositions (i.e., $D_{cri}$) evolve
with mixing and aging across particle populations. $D_{cri}$ demonstrates heightened
sensitivity to minor χ fluctuations at low mixing states (χ < 0.5; Fig. 6a), whereas further
increases in internal mixing (higher χ) exert negligible influence on $D_{cri}$ for already
internally mixed particles. This behavior suggests that the $D_{cri}$-χ relationship may
enable a novel parameterization for $D_{cri}$ estimation, a framework that is not yet reported



in prior literature.

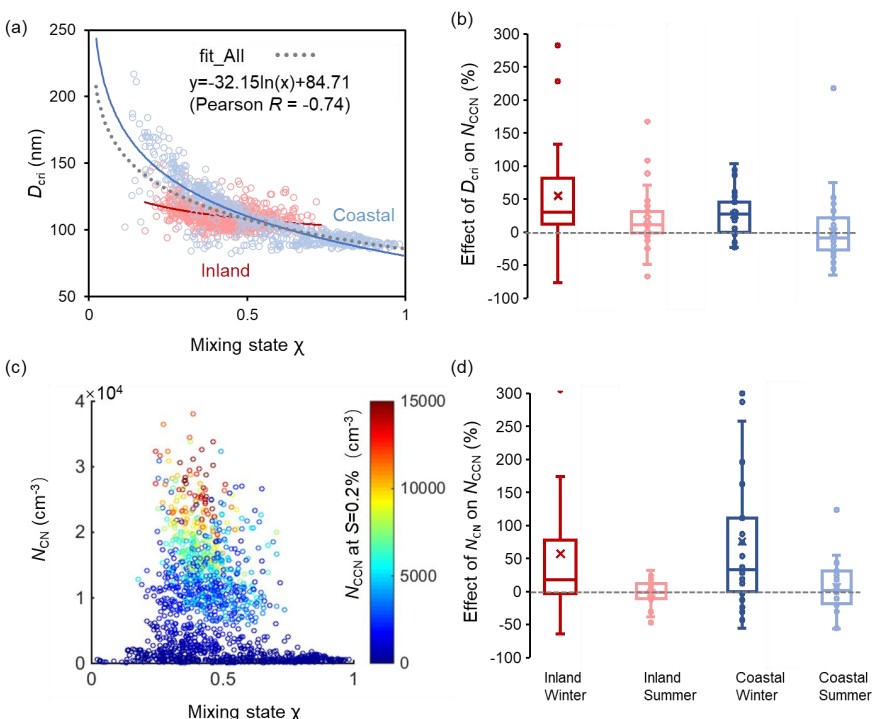

**Fig 6.** Dependency of the critical diameter ($D_{cri}$) on the $\chi$ (a), relative change of CCN

number concentration ($N_{CCN}$) at supersaturation $S = 0.2\%$ with the reduction in $D_{cri}$ (b);

Dependency of the CN number concentration ($N_{CN}$) on the $\chi$, different colors represent

the $N_{CCN}$ (c), relative change of $N_{CCN}$ with the change in $N_{CN}$ (d).

Coastal aerosol data points (blue dots) span a broad $D_{cri}$ range (80–220 nm) with
$\chi$ varying from 0.1 to 1, reflecting alternating influences of highly hygroscopic
inorganic salts (sea salt, sulfate) and less-hygroscopic organic matter. In contrast, inland
aerosols—dominated by anthropogenic pollutants—exhibit a narrower $D_{cri}$ range (90–
150 nm). Both environments show negative $D_{cri}$-$\chi$ correlations, but with distinct
functional forms: coastal aerosols feature an exceptional logarithmic fit ($D_{cri}$ = -



42.98ln($\chi$) +80.36, R² = 0.75; Fig. 6a blue line), while inland aerosols (red line) yield a
shallower slope (-12.04). Pooling all data, we derive a generalized parameterization:
$D_{cri}$ = -32.15ln($\chi$) + 84.71 (Pearson r = -0.74, R² = 0.54).
Box plot analyses (Fig. S6) show mixing state reduces $D_{cri}$ by 2.2–6.8% across
campaigns, with the steepest winter decline. $\chi$ impacts on $N_{CN}$ differ starkly between
environments: positive effects in polluted inland air (+9%) versus negative effects in
coastal regions (-2%). Inland aerosols, frequently perturbed by primary emissions and
new particle formation, exhibit elevated $N_{CN}$ (peaking at $\chi$ = 0.2–0.7), while coastal $N_{CN}$
remains ~5000 cm⁻³ across all $\chi$.
To isolate the impacts of critical diameter ($D_{cri}$) and condensation nuclei number
concentration ($N_{CN}$) on CCN activity, we categorized data into two groups: C1 (particles
within specific $N_{CN}$ ranges) evaluates $N_{CCN}$ variations driven by $D_{cri}$-$\chi$ relationships,
while C2 (particles within fixed $D_{cri}$ intervals) assesses $N_{CN}$-$\chi$ effects (Fig. 6b). Relative
changes (RC) in $D_{cri}$, $N_{CN}$, and $N_{CCN}$ with $\chi$ were calculated by comparing successive $\chi$
increments ($\chi$i+1 vs. $\chi$i, i=0,0.1…1) within defined $N_{CN}$/$D_{cri}$ windows.
Notably, $\chi$ exerts more pronounced effects on $N_{CCN}$ for externally mixed aerosols.
For example, coastal winter aerosols (high external mixing; $\chi_{mean}$=0.38±0.12) showed
$N_{CCN}$ RCs of 23% (C1) and 72% (C2), whereas coastal summer aerosols (high internal
mixing; $\chi_{mean}$=0.69±0.19) exhibited negligible effects (-2.5% in C1, 0.9% in C2). Inland
atmospheres, despite smaller seasonal $\chi$ variations, showed analogous trends: winter
$N_{CCN}$ RCs (55% in C1, 57% in C2 for external mixing) exceeded summer values for
more internally mixed populations (Fig. 6d). These results confirm that hygroscopic



heterogeneity strongly influences $N_{CCN}$ under external mixing, aligning with prior work
(Ching et al., 2017).

Mixing state impacts on $N_{CCN}$ are most pronounced during winter in both

environments, attributed to heightened winter $D_{cri}$ sensitivity to χ: a 0.1 χ increase
reduces $D_{cri}$ by 5.2% (winter), boosting $N_{CCN}$ by 39%, versus 2.4% $D_{cri}$ reduction
(summer) yielding only 6% $N_{CCN}$ enhancement. Concomitantly, winter $N_{CN}$-χ effects on
$N_{CCN}$ reach 65%, far exceeding summer responses.

Contrasting with prior evaluation methods that oversimplify mixing states (Ren et

al., 2018; Xu et al., 2021b), the entropy-based framework adopted herein enables
explicit quantification of CCN activity evolution in response to mixing state transitions.
Inland winter aerosols are presumably shaped by intense urban pollution sources—
including traffic emissions, residential heating, and cooking activities—thereby
enriching the externally mixed particle fraction (Fan et al., 2020; Xie et al., 2020).
Analogously, coastal winter aerosols exhibit dominant external mixing, consisting of
near-hydrophobic and hydrophilic particle mixtures (Xu et al., 2021a). As illustrated in
Fig. S2, winter aerosol populations display bimodal or multimodal $\kappa$–PDF distributions,
evidencing high-degree external mixing with chemically diverse compositions. These
results collectively highlight the pivotal role of mixing state heterogeneity in
modulating CCN activity across environments.
**4. Conclusions**

The mixing state of aerosol populations undergoes complex transformations

during atmospheric aging, altering the distribution of hygroscopic and non-hygroscopic



392 components and thus influencing CCN activity (Xu et al., 2021a; Ching et al., 2017).

393 This study derived a mixing state index from field-measured hygroscopicity probability

394 density functions, systematically investigating its impacts on CCN activity in inland

395 and coastal environments. Results provide field evidence that aerosol mixing states

396 generally reside between purely internal and external extremes (Chen et al., 2022),

397 highlighting a dual regulatory mechanism of mixing state on CCN activity. As $\chi$

398 increases, CN number concentrations ($N_{CN}$) first rise—driven by primary emissions and

399 new particle formation—then decline due to condensation and coagulation during aging.

400 Additionally, a logarithmic decreasing relationship between critical diameter ($D_{cri}$) and

401 $\chi$ was identified for both inland and coastal particles, parameterized as $D_{cri} = -32.15\ln(\chi)$

402 $+ 84.71$ (Pearson $R = -0.74$, $R^2 = 0.54$). This offers a practical approach to estimate $D_{cri}$

403 from $\chi$, serving as a general framework for integrating mixing state effects on CCN

404 activity in atmospheric models.

405  Entropy-based analyses confirm the pivotal role of mixing state in regulating $N_{CCN}$,

406 especially for externally mixed aerosols: a 0.1 $\chi$ increase can enhance $N_{CCN}$ by 39–65%.

407 Current models often oversimplify aerosol mixing states as purely internal or external

408 (Stevens et al., 2019; Bauer et al., 2013), the latter being particularly sensitive to organic

409 matter (Ren et al., 2018; Bhattu et al., 2015). Such simplifications introduce significant

410 biases in $N_{CCN}$ estimation (Riemer et al., 2019; Ching et al., 2019). The $\chi$-$D_{cri}$

411 parameterization proposed here offers a novel approach to reduce model complexity in

412 representing aerosol hygroscopicity and CCN activation, enabling more accurate

413 simulations of aerosol CCN capacity. This advancement improves our understanding of



aerosol-cloud interactions (IPCC, 2021; Rosenfeld et al., 2019), critical for refining
climate effect assessments.

### Data availability

All data used in the study are available at https://doi.org/10.3974/geodb.2019.06.11.V1
(Fan et al., 2019) and http://doi.org/10.17632/3dx6pnx869.1 (Xu et al., 2021a).

### Author contributions

RH and JR conceived the conceptual development of the paper. JR, FZ and WX directed
and performed the experiments with YW and LC. FZ provided the dataset in the inland
site. JO, DC and CO provided the dataset in the coastal site. JR conducted the data
analysis and wrote the draft. All authors edited and commented on the various sections
of the paper.

### Competing interests

The contact author has declared that none of the authors has any competing interests.

### Supporting Information

Additional analysis results that were applied in this study. Sensitivity of the
hygroscopic parameter for the group of the hygroscopic species on the mixing state
index $\chi$ (Figure S1), mean values of the $\kappa$–PDF for aerosols of five diameters (Figure
S2), time series of the average per-particle species diversity $D\alpha$, the bulk population
species diversity $D\gamma$, and their affine ratio $\chi$ (Figure S3), variation of the peak diameter
($D_{peak}$) with the mixing state index (Figure S4), diurnal variation of $\chi$ and CN



concentration during winter and summer periods for 40 nm and 150 nm aerosols in
inland and for 35 nm and 165 nm aerosols in coastal site (Figure S5), relative change
of the critical diameter and CN concentration with the mixing state index χ (Figure S6)
(PDF).
**Acknowledgements**
This work was funded by the National Natural Science Foundation of China (NSFC)
under Grant No. 42525301, 42405118 and 42475112, the State Key Laboratory of
Loess Science, Institute of Earth Environment, Chinese Academy of Sciences
(SKLLQG2429), the Guangdong Natural Science Foundation (Grant No.
2024A1515011005). We thank all participants of the field campaign for their hard work
and sharing of the data.

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
