# Peer review of "Contrasting Inland-Coastal Aerosol Mixing States: An Entropy-Based"

_EGUsphere, 2025_

## Author Comment (AC1)

**Referee #2**

This manuscript, titled "Contrasting Inland-Coastal Aerosol Mixing States: An Entropy-Based Metric for CCN Activity", presents a systematic investigation of how aerosol mixing states—specifically, the degree of internal vs. external mixing—affect cloud condensation nuclei (CCN) activity in two contrasting environments: inland (urban Beijing) and coastal (Mace Head, Ireland). This work bridges a critical gap between aerosol microphysics and climate modeling by providing a quantitative framework to incorporate realistic mixing state effects on CCN activity. It underscores the need to move beyond binary mixing assumptions and adopt entropy-based metrics for more accurate climate projections, particularly in diverse and dynamic environments like urban and coastal regions. This paper merits publication after minor revisions. To further strengthen this manuscript, the following questions should be addressed:

Why were these two locations chosen? Are they representative for inland and coastal? If possible, efforts should be made to collect more mixing state data from inland and coastal to support this discussion.

Re: Thank you for your suggestion. The IAP field measurements used in this study were collected during winter and summer as a part of the Atmospheric Pollution and Human Health in a Chinese Megacity (APHH-Beijing) program (Shi et al., 2019). The observation site is located between the Third and Fourth Ring Roads in Beijing, China. It is a typical urban site with significant aerosol population variability, primarily influenced by local anthropogenic sources such as vehicles, cooking emissions, and residential heating. In contrast, the MHD observation and research facility is situated on the west coast of Ireland. Observations indicate that over 60% of air masses at MHD are classified as clean oceanic air masses (O'Dowd et al., 2014), while the remaining 40% are subject to varying degrees of anthropogenic influence. The primary objective of this study is to investigate the heterogeneity of hygroscopicity of aerosol particles in polluted inland and clean coastal regions. For the first time, we apply the mixing state index based on entropy theory to real-world atmospheric conditions to explore its impact on cloud condensation nuclei (CCN) activity. To achieve this, we selected four datasets: IAP winter, IAP summer, MHD winter, and MHD summer.

Indeed, to facilitate a more comprehensive comparison between inland and coastal areas, it is essential to gather additional mixed state datasets. However, most reported mixed state indices currently available are based on $\chi$ calculated from chemical composition or standard deviation of the $\kappa$-PDF. A detailed dataset of hygroscopicity distribution is crucial for characterizing the heterogeneity of hygroscopicity in mixing state index. Unfortunately, such data are still relatively scarce. Therefore, we have revised the title as: "**Contrasting Aerosol Mixing States at Inland and Coastal Sites: An Entropy-Based Metric for CCN Activity**".

"But the current models lack regional-specific mixing state parameters and usually assume uniform mixing in both environments. This could lead to large uncertainties in predicting CCN concentrations, highlighting the need for site-specific observations." I suggest discussing in detail this substantial uncertainty and its sources.

Re: The sentence has been revised as follows or **Lines 88-102**: "…However, the current models lack regional-specific mixing state parameters and usually assume uniform mixing in both environments. This could lead to large uncertainties in predicting CCN concentrations, highlighting the need for site-specific observations. For example, Ren et al. (2018) found that the impact of aerosol mixing state on CCN activation characteristics ranged from -34% to +16 % in urban atmosphere. Comparison between a fully internal mixture assumption and using the mixing state index from the particle-resolved model, Ching et al. (2017) found the obvious overestimation in CCN concentration estimation. Especially in the regions eg., Amazon Basin, Central Africa and Indonesia, the particles appeared to be more external, errors in CCN concentration would increase up to 100% (Hughes et al., 2018). A detailed exploration of mixing state on CCN concentration in global scale was conducted by Zheng et al. (2021a), and the results showed that the mixing state varied spatially with more externally mixed over the North Atlantic Ocean, off the coasts of Southern Africa, and Australia. Thus, assuming particles with internally-mixed would introduce errors in CCN concentration of 50-100%…"

Does the surrogate choice (κNH = 0.01, κH = 0.6–0.8) fully capture the hygroscopic diversity of organics, especially oxygenated and fresh POA?

Re: Thanks for your suggestion, here the heterogeneity in aerosol hygroscopicity is calculated based on the measurement of κ-PDF, differing from previous reported χ based on the chemical diversity (Ching et al., 2017; Zheng et al., 2021) by grouping two surrogate species (one with BC and POA, the other with inorganic and secondary organic aerosol species). By referred from Yuan et al. 2021, the key assumption is that the aerosol containing aerosol particles is a binary system consisting of the non- and/or less hygroscopic ($\kappa_N$ of <0.05) and more hygroscopic components ($\kappa_H$ of 0.5-0.6, referred inorganics), which corresponds to the minimum and maximum hygroscopic parameters. In ambient atmosphere, each aerosol particle in the population contains one or two of the components. As shown in Figure R1 or S1, κ-PDF at IAP can be considered the normalized aerosol number fractions varied with κ between 0 and 0.6. and at MHD atmosphere, κ varied between 0 and 0.8. Thus, considering of the variation in $\kappa_H$, calculation assuming $\kappa_H$ of 0.6 and 0.8 given in Figure S2 and the results show that the mixing state index do not differ significantly from those calculated assuming $\kappa_H$ of 0.6. So, the calculation assumes $\kappa_H$ of 0.6 was chosen in our study. The sensitivity of the hygroscopic parameter for the group of the hygroscopic species on the mixing state index χ was done both for the inland and coastal aerosols as seen in the revised text. See follows and **Lines 193-204**:

"…To characterize the heterogeneous distribution of the hygroscopic and non-hygroscopic components in populations (Chen et al., 2022b), we calculated the mixing state index (χ) using the $\kappa$-PDF, following the methodology of Yuan et al. (2023). Two surrogate groups in a population of $N$ aerosol particles were assumed (Zheng et al., 2021a). One surrogate group consists the non- and/or slightly hygroscopic species with $\kappa_N$ of <0.05 and another group contains the more hygroscopic species with $\kappa_H$ of 0.5-0.6 (Yuan et al., 2023, referred inorganics). Ambient particles typically contain one or two of the components and the $\kappa$ lies between 0 and 0.6 at IAP or 0.8 at MHD as shown

in Figure S1. Taking into account the enhanced hydrophilicity of marine aerosols at MHD site, calculation assuming $\kappa_H$ values of 0.7 and 0.8 were shown in Fig. S2. While these variations in $\kappa_H$ introduced a mean uncertainty of 8% in $\chi$ values, it did not significantly affect the seasonal or site comparisons ..."

[Figure]

**Fig. R1 and Fig. S1** Mean value of the $\kappa$–PDF for aerosols of five diameters during winter and summer periods at IAP (a and b) and MHD (c and d) sites.

Do the short campaign windows ($\approx$ 1 month per season) adequately represent inter-annual variability in air-mass type and photochemical intensity?

Re: Although the short campaign windows might not represent inter-annual variability in air-mass type and photochemical intensity well, it provided direct observational evidence that the current model does not consider the spatial differences of mixing states, which leads to a significant overestimation of CCN concentration. In addition, to investigate the potential impact of mixing states on CCN concentration, simultaneous observation of particle size distribution, hygroscopic distribution, chemical composition and CCN is required. The field observations of IAP and MHD sites provide us with the feasibility to explore the spatial differences between inland and coastal mixing states in this study. Of course, it is necessary to conduct long-term observations in the future to investigate the temporal variations of mixing states.

Some statements have been added as follows or See **Line 123-137**: "…The inland atmospheric measurements were conducted for two campaigns from 16 November to 6 December 2016 and 29 May to 13 June 2017 as a part of the Air Pollution and Human Health (APHH) project (Shi et al., 2019), at the Institute of Atmospheric Physics, Chinese Academy of Sciences (IAP, 39.97° N, 116.37° E) in urban Beijing. The campaigns were complemented by the hygroscopicity and CCN observations and were conducive to provide information on the aerosol hygroscopicity affecting urban pollutions. This urban site exhibited highly variable aerosol populations dominated by local anthropogenic sources including vehicular, cooking emissions, and residential heating. Coastal measurements were performed at the Mace Head atmospheric research station (MHD, 53.33° N, 9.90° W) from 1 November 2009 to 30 January 2010, and summer periods from 11 to 31 August 2009 and July 2010, which located on the west

coast of Ireland. Aerosol particles here experience alternating influences from polluted continental and clean marine atmospheres. The map of the sites was shown in Figure 1. More details about the campaigns were given in Fan et al. (2020) and Xu et al. (2021a) …"

Is there evidence of κ-köhler non-ideality that would invalidate the single-parameter κ assumption at high S?

Re: The single-parameter κ assumption may not be valid under high supersaturation (S) conditions due to non-ideal behavior in surface tension and bulk composition. For example, surfactants in aerosol can lower surface tension (Ovadnevaite et al., 2017; Fan et al., 2024), which is not considered in the classical Köhler theory. Additionally, interactions in multi-component aerosol systems can also lead to non-ideal behavior. Experimental studies have shown deviations from the $\kappa$ assumption under high S conditions. Therefore, more complex models are needed to accurately describe aerosol activation. This study is mainly focused on exploring the possible impact of mixing state on CCN concentration at the supersaturation of ~0.2% as an example, due to the less sensitivity to mixing state at high S (Bhattu et al., 2015).

How about the results if the proposed parameterization being implemented in a sectional or modal aerosol model to quantify the reduction in CCN bias compared to default internal/external mixing assumptions?

Re: Thanks for the suggestion, some statements have been added as follows or See **Line 557-570**:

"…Entropy-based analyses confirm the pivotal role of mixing state in regulating $N_{CCN}$, especially for externally mixed aerosols: a 0.1 χ increase can enhance $N_{CCN}$ by 39–65%. Current models often oversimplify aerosol mixing states as purely internal or external (Stevens et al., 2019; Bauer et al., 2013), the latter being particularly sensitive to organic matter (Ren et al., 2018; Bhattu et al., 2015). Such simplifications introduce significant biases in $N_{CCN}$ estimation (Riemer et al., 2019; Ching et al., 2019). The $\chi$-$D_{cri}$ parameterization proposed here offers a novel approach to reduce model complexity in representing aerosol hygroscopicity and CCN activation, enabling more accurate simulations of aerosol CCN capacity. It is expected mitigate the underestimation in CCN compared with the complete external mixing assumption, while effectively alleviates the overestimation that arises from applying the complete internal mixing assumption in regions characterized by high external mixing (Zheng et al., 2021a). This advancement improves our understanding of aerosol-cloud interactions (IPCC, 2021; Rosenfeld et al., 2019), critical for refining climate effect assessments."

What is the sensitivity of the Pearson correlation (r = –0.74) to random vs. systematic errors in Dcri?

Re: Here we use simulated data to evaluate the sensitivity of Pearson correlation coefficient to random error ϵ and systematic error δ in $D_{cri}$. As shown in the **Fig. R2**, the correlation coefficient is not sensitive to systematic errors. Further introducing the random error ϵ (standard deviation σ ranging from 1.5 to 15), as ϵ increases, the correlation coefficient decreases from 0.74 to 0.52, indicating correlation coefficient is more sensitivity to random error in $D_{cri}$.

[Figure]

**Fig. R2** Dependency of the critical diameter on the χ, with different systematic errors. The relationship between Dcri and χ is heavily influenced by the chemical composition of particulate matter, as well as factors such as new particle formation, emission sources, and secondary reactions. Consequently, this relationship may exhibit significant variations across different regions. Is it possible for the author to further verify this using the results directly from chemical composition analysis?

[Figure]

**Fig. R3 and Fig. 5** Case in IAP-winter and IAP-summer. Particle number size distribution and PM1 (a), mass fraction of the PM$_1$ and the critical diameter (b), mixing state index (χ), number fraction of the nearly hydrophobic mode (NH) and more hygroscopic mode (MH) for 40 nm particles (c), χ, NH and MH for 150 nm particles

(d).

[Figure]

**Fig. R4 and Fig. 6** Case in MHD-winter and MHD-summer. Particle number size distribution and PM1 (a), mass fraction of the $PM_1$ and the critical diameter (b), mixing state index ($\chi$), number fraction of the nearly hydrophobic mode (NH) and more hygroscopic mode (MH) for 35 nm particles (c), $\chi$, NH and MH for 165 nm particles (d).

Re: Thanks for the suggestion, some statements have been added as follows or See **Line 483-496**:

"…As already discussed above, strong impact of primary emission and secondary formation on aerosol mixing state was observed in both sites (Fig. 5 and 6). It also provides even more details on the $D_{cri}$-$\chi$ correlations. For example, the $D_{cri}$ exhibited rapidly increased with the primary emissions (ie., mass fraction of POA enhanced) during polluted periods. The $D_{cri}$ pattern appeared opposite with that of the mixing state index, especially for the accumulation-mode particles. More pronounced $D_{cri}$-$\chi$ correlations were observed during the new particle formation (Fig. 5a1-d1). The decreasing presence of $D_{cri}$ matched the increasing proportion of $SO_4^{2-}$ and SOA with the $\chi$ increased during NPF events. Similar correlations between the critical diameter and mixing state index were also found in the coastal atmosphere, especially for the case of the enhanced anthropogenic organic matter and sea salt production (Fig.6). This implies that the relationship between the $D_{cri}$ and $\chi$ might be disturbed by the variation of emission pollution and secondary formation processes, resulting in spatiotemporal differences…"

The inland atmospheric measurements were conducted for two periods from 16

November to 6 December and 29 May to 13 June" in which year?

Re: Revised and See **Lines 123-127**: "…The inland atmospheric measurements were conducted for two campaigns from 16 November to 6 December 2016 and 29 May to 13 June 2017 as a part of the Air Pollution and Human Health (APHH) project (Shi et al., 2019), at the Institute of Atmospheric Physics, Chinese Academy of Sciences (IAP, 39.97° N, 116.37° E) in urban Beijing ..."

**References for the Response**

Shi, Z., Vu, T., Kotthaus, S., Harrison, R. M., Grimmond, S., Yue, S., Zhu, T., Lee, J., Han, Y., Demuzere, M., Dunmore, R. E., Ren, L., Liu, D., Wang, Y., Wild, O., Allan, J., Acton, W. J., Barlow, J., Barratt, B., Beddows, D., Bloss, W. J., Calzolai, G., Carruthers, D., Carslaw, D. C., Chan, Q., Chatzidiakou, L., Chen, Y., Crilley, L., Coe, H., Dai, T., Doherty, R., Duan, F., Fu, P., Ge, B., Ge, M., Guan, D., Hamilton, J. F., He, K., Heal, M., Heard, D., Hewitt, C. N., Hollaway, M., Hu, M., Ji, D., Jiang, X., Jones, R., Kalberer, M., Kelly, F. J., Kramer, L., Langford, B., Lin, C., Lewis, A. C., Li, J., Li, W., Liu, H., Liu, J., Loh, M., Lu, K., Lucarelli, F., Mann, G., McFiggans, G., Miller, M. R., Mills, G., Monk, P., Nemitz, E., O'Connor, F., Ouyang, B., Palmer, P. I., Percival, C., Popoola, O., Reeves, C., Rickard, A. R., Shao, L., Shi, G., Spracklen, D., Stevenson, D., Sun, Y., Sun, Z., Tao, S., Tong, S., Wang, Q., Wang, W., Wang, X., Wang, X., Wang, Z., Wei, L., Whalley, L., Wu, X., Wu, Z., Xie, P., Yang, F., Zhang, Q., Zhang, Y., Zhang, Y., and Zheng, M.: Introduction to the special issue "In-depth study of air pollution sources and processes within Beijing and its surrounding region (APHH-Beijing)", Atmos. Chem. Phys., 19, 7519–7546, https://doi.org/10.5194/acp-19-7519-2019, 2019.

O'Dowd, C., Ceburnis, D., Ovadnevaite, J., Vaishya, A., Rinaldi, M., & Facchini, M. C.: Do anthropogenic, continental or coastal aerosol sources impact on a marine aerosol signature at Mace Head?, Atmos. Chem. Phys., 14(19), 10687–10704, https://doi.org/10.5194/acp-14-10687-2014, 2014.

Bhattu, D., Tripathi, S. N.: CCN closure study: Effects of aerosol chemical composition and mixing state, Journal of Geophysical Research: Atmospheres., 120(2), 766-783, https://doi.org/10.1002/2014JD021978, 2015.

Ching, J., Fast, J., West, M., Riemer, N.: Metrics to quantify the importance of mixing state for CCN activity, Atmospheric Chemistry and Physics., 17(12), 7445-7458, https://doi.org/10.5194/acp-17-7445-2017, 2017.

Zheng, Z., Curtis, J. H., Yao, Y., Gasparik, J. T., Anantharaj, V. G., Zhao, L., West, M., Riemer, N.: Estimating submicron aerosol mixing state at the global scale with machine learning and Earth system modeling, Earth and Space Science., 8(2), e2020EA001500, https://doi.org/10.1029/2020EA001500, 2021.

Ovadnevaite, J.; Zuend, A.; Laaksonen, A.; Sanchez, K. J.; Roberts, G.; Ceburnis, D.; Decesari, S.; Rinaldi, M.; Hodas, N.; Facchini, M.; Seinfeld, J. H.; O' Dowd, C.: Surface tension prevails over solute effect in organic-influenced cloud droplet activation, Nature, 546, 637−641, 2017.

Fan, T., Ren, J., Liu, C., Li, Z., Liu, J., Sun, Y., Wang, Y., Jin, X., Zhang, F.: Evidence of Surface-Tension Lowering of Atmospheric Aerosols by Organics from Field Observations in an Urban Atmosphere: Relation to Particle Size and Chemical Composition, Environ. Sci. Technol, 58, 26, 11363–11375, https://doi.org/10.1021/acs.est.4c03141, 2024.

---

## Author Comment (AC2)

**Referee #1**

- **General comments**:

This study aims to investigate aerosol mixing state characteristics at two sites—one inland and one coastal—and to examine their impacts on CCN activity. The comparisons and coastal measurements presented here indeed reveal some interesting features that could potentially offer new insights. However, the current manuscript contains numerous misleading statements and conclusions, as detailed in my major and specific comments.

The most important concern is that the key findings rest on conceptual flaws that could mislead less-experienced readers, such as students. Before explaining the reasons, the definition of "mixing state" must be clarified.

If the authors are quantifying the mixing state index for the *entire aerosol population*, the term refers to the distribution of aerosol mass among different particles of different sizes, under the condition of fixed bulk mass fractions of aerosol components. This implies that overall aerosol hygroscopicity is fixed (assuming the volume mixing rule applies).

If, instead, the authors are quantifying the mixing state index for aerosols of *different diameters*, the term refers to the distribution of aerosol mass among particles under fixed aerosol size distribution and size-resolved chemical composition.

In models, when both aerosol size distribution and size-resolved chemical composition are known, the mixing state refers to how aerosol mass is distributed across particles of the same size, corresponding to $\chi$ variations of different sizes under fixed size distributions and size-resolved compositions. Thus, when discussing the impact of mixing state, it should be done under fixed aerosol size distributions and bulk aerosol compositions (or ideally, fixed size-resolved compositions).

With this in mind, my following points are justified. Theoretically, CCN activity depends primarily on aerosol size distribution, with hygroscopicity playing a secondary role. Moreover, the role of *mixing state*—could be represented by the hygroscopicity distribution—is generally smaller than that of the overall hygroscopicity. In this study, the authors parameterize the critical activation diameter ($D_{(cri)}$), which is determined by aerosol hygroscopicity and supersaturation, with the mixing state index $\chi$. The reported relationship between $D_{(cri)}$ and $\chi$ does not isolate the impact of mixing state; it simply reflects their co-variation. Higher $\chi$ often corresponds to higher internal mixing, which is mainly driven by secondary aerosol formation and thus higher hygroscopicity. Therefore, $\chi$ can correlate with $D_{(cri)}$, however, without being its causal driver.

Re: We are extremely grateful for the comments and suggestions you have provided regarding our manuscript. After carefully considered each of your points, we have made the corresponding revisions to address them and we believe that these revisions have significantly strengthened our paper. Current models often oversimplify mixing states by assuming pure internal or external mixing (Stevens et al., 2019; Riemer et al., 2019; Zheng et al., 2021) and further estimates the critical diameter based on the $\kappa$-Köhler theory (Petters and Kreidenweis, 2007). This would introduce significant uncertainty in

the evaluation of cloud condensation nuclei (CCN) activity. In this study, for understanding of the mixing state variations and its impact on CCN activity quantitatively, the mixing state entropy ($\chi$) proposed by Riemer and West was applied by combining the probability distribution function (PDF) of $\kappa$ from the in-situ measurement of the HTDMA at two Inland and Coastal Sites. This metrics describe ambient particle mixing states more reasonably and have been applied to quantify the impact of mixing state on various quantities relevant to climate and air quality (Ching et al., 2017; 2019; Zheng et al., 2021; Zhao et al., 2021). Although there have been studies evaluating the $\chi$ in models (ie., particle-resolved model, GCMs, ESM and so on), its parameterization relationship with CCN has not yet been established. Furthermore, the mixing state index is devised mostly based on the chemical diversity concerning the distribution of chemical composition across the aerosol population. $\chi$ index from previous field studies were mainly based on chemical composition, which relative biased towards larger sizes. On the other hand, HTDMA provided valuable insights into the aerosol mixing state for sub-micron particles, which directly related to CCN budget. Therefore, this study aimed to fill the gap by investigating the impact of heterogeneity in aerosol hygroscopicity on CCN activity and its relevant climate effects. Furthermore, by parameterizing the mixing index to $\chi$, the results are more comparable to previous studies (Ching et al., 2017).

[Figure]

**Fig. R1** The concept of the aerosol mixing state index for a population consisting of two species, where the $\kappa$-PDF for five diameters is used for calculation $\chi$.

Given these considerations, and with reference to Yuan et al. (2023), the mixing state index $\chi$ proposed in this study is devised based on the concept of information entropy. Although this parameter is not yet fully quantified and its consideration of particle size is limited, it can effectively describe the heterogeneity of hygroscopicity across the aerosol population. Therefore, we attempted to parameterize the critical diameter based on this index to help reduce the uncertainty of CCN and its climate effects in model. In this study, the $\kappa$-PDF for 40, 80, 110, 150 and 200 nm at IAP site and 35, 50, 75, 110 and 165 nm at MHD site is used for calculating the $\chi$. The key assumption is that an aerosol containing $N$ particles is a binary system. One surrogate group consists the non- and/or slightly hygroscopic species with $\kappa_N$ of <0.05 and another group contains the more hygroscopic species with $\kappa_H$ of 0.5-0.6 (Yuan et al., 2023, referred inorganics). Each aerosol particle in the population contains one or two of the components. Mixing state index $\chi$ of 0 and 1 indicate completely externally- and

completely internally-mixed aerosol population, respectively. Most ambient particle populations have $\chi$ intermediate between 0 and 1.

Detailed introduction of the mixing state index has been added in the revised text for clarity. See follows and **Lines 103-120**:

"…Therefore, for quantifying the aerosol mixing state in the ambient atmosphere, we apply the algorithm of entropy proposed by Riemer and West (2013) to investigate the aerosol heterogeneity. This index has been applied to quantify the mixing state more reasonably both in field campaigns (Zhao et al., 2021; Yuan et al., 2023) and model simulations (Ching et al., 2016; Zheng et al., 2021a). However, most studies focused on quantifying the particle heterogeneity in composition (Ching et al., 2019; Fierce et al., 2020; Zhao et al., 2021). Here we concentrated on evaluating the heterogeneity in aerosol hygroscopicity for sub-micron particles, which directly related to CCN budget. By refereeing to Yuan et al. 2023, the heterogeneity in hygroscopicity was investigated by combining in-situ measurements of probability distribution function of the hygroscopicity with the algorithm of entropy. Briefly, the mixing state index $\chi$, is devised based on the concept of information entropy concerning the distribution of hygroscopicity across the aerosol population. It varies between 0 (external mixing completely) and 1 (internal mixing completely). By integrating inland and coastal measurements, this study will focus on addressing two key gaps, (1) How continental vs. marine-dominated environments shape aerosol mixing states and CCN activity; (2) Whether $\chi$-based CCN parameterizations show regional dependencies, providing critical constraints for climate models ..."

To genuinely examine the impact of mixing state on CCN activity, the authors should test scenarios with fixed aerosol size distribution and fixed aerosol composition (so that overall $\kappa$ is unchanged). Variations in $D_{(cri)}$ could then be explored under different combinations of $\kappa$ (derived from composition measurements) and $\chi$. A 2-D plot with $\kappa$ on the x-axis and $\chi$ at ~150 nm on the y-axis, followed by analysis of $D_{(cri)}$ variations, could yield more robust insights.

Re: Thank you very much for your suggestion. We would like to clarify that the original intention of this study was to consider that the current model's estimation of critical activation particle size is still based on the assumption of complete internal mixing and external mixing (Riemer et al., 2019), which makes the characterization of CCN activity highly uncertain. Although there have been attempts to use mixing state parameters to evaluate their impact on CCN concentration, a parametric relationship has not yet been established. In this study, the mixing state index $\chi$ mainly reflects the heterogeneity of hygroscopicity distribution (calculated based on $\kappa$-PDF). We noticed that particle size distribution and chemical composition change correspondingly with changes in $\chi$ with the evolution of the emissions and aging processes, as shown in Figure R2 or Figure 7.

As $\chi$ increases, the peak diameter ($D_{peak}$) of the particle size distribution (PNSD) shifts towards larger sizes (Figure 7a and Figure S7), while the peak concentration appears in the middle $\chi$ range (0.3-0.6). This trend indicates that driven by primary emissions and the formation of new particles, the CN number concentration ($N_{CN}$) first increases and then decreases due to mixing and aging processes (Figure 7b).

The critical diameter ($D_{cri}$) - defined as the minimum size activated at a given

supersaturation - depends on the hygroscopicity of the aerosol. This hygroscopicity is determined by the mass fraction of hygroscopicity and soluble components (Petters and Kreidenweis, 2007). Taking the measurement at 0.2% supersaturation as an example, Figure 7c shows that $D_{cri}$ decreases with the increase of highly hygroscopic inorganic components (such as sulfates and nitrates) in the inland atmosphere. In contrast, coastal $D_{cri}$ exhibits a non-linear variation of χ: high external mixing (low χ) increases $D_{cri}$ due to the main organic components and reduces sea salt particle fraction. With the increase of χ, the mass fraction of non- sea salt sulfate (nss sulfate) increases, and the activation potential is increased by reducing $D_{cri}$.

[Figure]

**Fig. R2 or Fig 7.** Comparison of the average particle number size distribution (PNSD) in different mixing state index (χ) (a), CN number concentration ($N_{CN}$) as a function of χ (b), Critical diameter ($D_{cri}$) at S=0.2% and mass fraction of chemical composition as a function of χ (c), CCN number concentration ($N_{CCN}$) (d) and activation ratio (AR) at S=0.2% a function of χ (e).

Different from previous studies based on mixing state assumptions (Yang et al., 2012; Ren et al., 2018), this study focuses on the two main factors affecting CCN concentration mentioned above (particle size distribution and composition). Firstly, it discusses how particle size and chemical composition change during the evolution of χ (with χ ranging from 0 to 1 and a step size of 0.1) (Figure 7). On this basis, the corresponding changes in CCN concentration were further quantified. To isolate the impacts of critical diameter ($D_{cri}$) and condensation nuclei number concentration ($N_{CN}$) on CCN activity, we categorized data into two groups: C1 (particles within specific $N_{CN}$ ranges) evaluates $N_{CCN}$ variations driven by $D_{cri}$-χ relationships, while C2 (particles within fixed $D_{cri}$ intervals) assesses $N_{CN}$-χ effects. Relative changes (RC) in $D_{cri}$, $N_{CN}$, and $N_{CCN}$ with χ were calculated by comparing successive χ increments (χi+1 vs. χi, i=0,0.1…1) within defined $N_{CN}/D_{cri}$ windows.

Statements such as:
"Mixing state impacts on $N_{(CCN)}$ are most pronounced during winter in both environments, attributed to heightened winter $D_{(cri)}$ sensitivity to χ: a 0.1 χ increase reduces $D_{(cri)}$ by 5.2% (winter), boosting $N_{(CCN)}$ by 39%, versus 2.4% $D_{(cri)}$ reduction (summer) yielding only 6% $N_{(CCN)}$ enhancement. Concomitantly, winter $N_{(CN)}$–χ

effects on $N_{(CCN)}$ reach 65%, far exceeding summer responses."

should be revised. Correlations only indicate co-variations and cannot be directly interpreted as causal impacts of χ on CCN.

Re: Here, to isolate the impacts of mixing state on CCN concentration, we first explored the change in CN concentration (particle size) and $D_{cri}$ (chemical composition or hygroscopicity) with the variation of χ. On this basis, the change in $N_{CCN}$ through these two ways were explored. We categorized data into two groups: C1 (particles within specific $N_{CN}$ ranges) evaluates $N_{CCN}$ variations driven by $D_{cri}$-χ relationships, while C2 (particles within fixed $D_{cri}$ intervals) assesses $N_{CN}$-χ effects (Fig. R2 or Fig. 7b). Relative changes (RC) in $D_{cri}$, $N_{CN}$, and $N_{CCN}$ with χ were calculated by comparing successive χ increments (χi+1 vs. χi, i=0,0.1…1) within defined $N_{CN}$ / $D_{cri}$ windows.

Revised as follows and See **Lines 518-522**: "…With the variation in mixing state index χ, changes in $N_{CCN}$ are most pronounced during winter in both environments, attributed to heightened winter $D_{cri}$ sensitivity to χ: a 0.1 χ increase reduces $D_{cri}$ by 5.2% (winter), boosting $N_{CCN}$ by 39%, versus 2.4% $D_{cri}$ reduction (summer) yielding only 6% $N_{CCN}$ enhancement. Concomitantly, winter $N_{CN}$-χ effects on $N_{CCN}$ reach 65%, far exceeding summer responses…"

In summary, the current findings do not provide critical constraints for parameterizing fine-aerosol CCN activity in climate models, nor do they reduce uncertainties in aerosol–climate effect estimates. The discussion does not bring genuinely new insights into CCN parameterization. I recommend major revisions, with a stronger focus on how secondary aerosol formation affects hygroscopicity distribution and thus χ. Section 3.1 and 3.2 analyses are preliminary. I suggest revising:

- ○ Section 3.2 → "Impacts of Primary Aerosol Emissions and Secondary Aerosol Formation on Aerosol Mixing State"
- ○ Section 3.3 → "Impact of Mixing State on CCN Activity" (with the role of mixing state isolated as described above)

Re: Thank you for your suggestion. We have added an analysis to section 3.2, revised as follows and See **Lines 339-408**: "

**3.2 Impacts of Primary Aerosol Emissions and Secondary Aerosol Formation on Aerosol Mixing State**

As already noted above, changes in χ were clearly associated with the chemical composition varying with site and season. The relationships between the mixing state index and the number fraction of hydrophobic and hygroscopic mode during four campaigns are presented in Figure S6. The χ exhibited negative correlations with the fraction of hydrophobic mode but a positive relationship with the fraction of hygroscopic particles, highlighting the markedly different effects of the primary emissions and secondary formation on aerosol mixing state (Tao et al., 2024). To gain more insight on this effect between inland and coastal atmosphere, four case are analyzed (Fig. 5 and 6): case for IAP-winter, case for IAP-summer, case for MHD-winter and case for MHD-summer.

Case for IAP-winter is a heavy polluted event with the mean PM mass concentration increased from 22 to 437 μg m$^{-3}$ (Fig. 5a-d). The 40- and 150-nm χ patterns shifted quickly during the pollution periods. With the mass fraction of

hydrophobic compounds (ie., POA) in PM$_1$ increased, the $\chi$ of 40-nm particles decreased from 0.5 to 0.2, that is, an enhanced NH mode and a weaken MH mode (Fig. 5b-c). At this stage, large particles for 150 nm are mainly from aqueous formation with more proportion of nitrate. The corresponding $\chi$ of 150 nm was higher. While with that the mass fractions of secondary organic and inorganic compositions increased, particles were more internal mixed with $\chi$ increased to be 0.6 for 40-nm and 0.53 for 150-nm particles.

[Figure]

**Fig. R3 or Fig 5.** Case in IAP-winter and IAP-summer. Particle number size distribution and PM$_1$ (a), mass fraction of the PM$_1$ and the critical diameter ($D_{cri}$) (b), mixing state index ($\chi$), number fraction of the nearly hydrophobic mode (NH) and more hygroscopic mode (MH) for 40 nm particles (c), $\chi$, NH and MH for 150 nm particles (d).

Case for IAP-summer is the typical new particle formation events (NPF) with the mean PM$_1$ of 13 µg m$^{-3}$ (Fig. 5a1-d1). With the evolution of NPF events, the $\chi$ of 40- and 150-nm particles increased to be 0.95 and 0.61 with the enhanced proportion of more-hygroscopic components (ie., SOA, NO$_3^-$, SO$_4^{2-}$). The $\chi$ pattern is opposite of that of number fraction of NH mode and consistent with the variation of MH mode (Fig. S6). Note that a sudden decrease in $\chi$ on June 11th was disturbed by the strong primary emission. The chemical mass fractions showed more POA and black carbon with an enhanced NH mode and a weaker MH mode (Fig. 5b1-d1). The $\chi$ of 40-nm particles decreased to be 0.4 and that of the 150-nm particles decreased to be 0.2. The $\chi$ patterns appear to similar transitions for Aitken and accumulation-mode particles during haze and NPF events. The increase in $\chi$ is synchronous with the increase in MH mode from

secondary formation but opposite with that of LH mode from primary emissions. This implies that the primary emissions would lead particles more external mixing while secondary formation would promote aerosol more internal mixed in Inland atmosphere.

[Figure]

**Fig. R4 or Fig 6.** Case in MHD-winter and MHD-summer. Particle number size distribution and PM$_1$ (a), mass fraction of the PM$_1$ and the critical diameter (b), mixing state index ($\chi$), number fraction of the nearly hydrophobic mode (NH) and more hygroscopic mode (MH) for 35 nm particles (c), $\chi$, NH and MH for 165 nm particles (d).

Case for MHD-winter is a high organic matter pollution event with the mean PM$_1$ of 5.2 μg m$^{-3}$ and 52% mass fraction of organics (Fig. 6a-d). Larger presence of anthropogenic organic matter resulted the NH mode for 35-nm particles to be 95% and 165-nm particles to be 53% (Fig. 6). The $\chi$ of 35- and 165-nm particles decreased with the NH mode increased (Fig. S6), similar with the case for IAP site. There was a steady increase in $\chi$ when the MH-mode particles started increasing with the increase in mass fraction inorganics, eg., 35 nm particles showed the mean $\chi$ increasing from 0.43 to 0.57 and 165 nm particles from 0.35 to 0.6. This indicated that the trend of aerosol mixing state closely followed the evolution emission and secondary formation.

Case for MHD-summer is an extremely clean event with the mean PM$_1$ of 0.7 μg m$^{-3}$ (Fig. 6a1-d1). The dominated MH mode was found throughout the case, which could be attributed from the high mass fraction of nss-sulfate (41% average). Compared with the case in MHD-winter, the mean proportion of organic has decreased to be 15%. Therefore, the $\chi$ remains at a high value (mean $\chi$ of 0.9 for 35-nm and 0.8 for 165-nm

particles). Until August 28th, a stronger increase in the mass fraction of sea salt and accordingly SS mode in larger-size particles was observed. The χ decreased rapidly with the decrease in MH mode and enhanced SS mode, especially for the accumulation mode particles, suggesting the sea spray production makes particles more externally mixed.

In summary, these results suggest that the primary emission and secondary formation drive the hygroscopic distribution and can result in significant variation of aerosol mixing state χ both in Inland and coastal atmosphere. The pattern of χ varied among site and season, highlighting the importance of considering the impact of mixing state on CCN activity."

**Major Comments**

1. **Title and site representation** – Can the inland–coastal contrast be represented solely by observations at two sites? Aerosol aging differs substantially across different continental and marine locations. Observations from two inland or two coastal sites could also yield contrasting characteristics. I Suggest consider a more neutral title, e.g., *"Contrasting Aerosol Mixing States at Two Inland and Coastal Sites: An Entropy-Based Metric for CCN Activity"*. In the text, I suggest using "IAP–winter" instead of "inland–winter" and "MH–winter" instead of "coastal–winter," while placing broader inland–coastal implications in the conclusion.

Re: Thanks for the comments, the title has been revised as "Contrasting Aerosol Mixing States at Inland and Coastal Sites: An Entropy-Based Metric for CCN Activity". The "Inland-winter" has been revised as "IAP-winter", "Inland-summer" has been revised as "IAP-summer", "Coastal-winter" has been revised as "MHD-winter", "Coastal-summer" has been revised as "MHD-summer".

2. **Sampling period** – The short observation period, especially the two-week inland summer dataset, weakens the robustness of the "contrasting" conclusion. For example, Beijing aerosol properties in June may differ significantly from those in August. The representativeness of these periods should be at least discussed.

Re: As a part of the Atmospheric Pollution and Human Health in a Chinese Megacity (APHH-Beijing) program, campaigns in 2016 winter and 2017 summer provide observations of the atmospheric chemistry and physics and help to understand atmospheric processes affecting urban air pollutants (Shi et al., 2019). We have revised the sentences as follows and See **Lines 123-131:**

"…The inland atmospheric measurements were conducted for two campaigns from 16 November to 6 December 2016 and 29 May to 13 June 2017 as a part of the Air Pollution and Human Health (APHH) project (Shi et al., 2019), at the Institute of Atmospheric Physics, Chinese Academy of Sciences (IAP, 39.97° N, 116.37° E) in urban Beijing. The campaigns were complemented by the hygroscopicity and CCN observations and were conducive to provide information on the aerosol hygroscopicity affecting urban pollutions. This urban site exhibited highly variable aerosol populations dominated by local anthropogenic sources including vehicular, cooking emissions, and residential heating…"

o **Relationship between hygroscopicity distribution, mixing state, and CCN activity** – This needs to be explained more clearly. A straightforward logical

framework would help (on the basis of my comments on mixing state at the very beginning). Actually, for DMA-CCN measurements, the size-revied AR could be fitted using the formula proposed by Rose et al. (2008) which have three key parameters including the Da (critical activation diameter), MAF (Maximum Activation fraction) and the heterogeneity parameter σ. Among three parameters, σ is mostly affected by the mixing state, or so-called heterogeneity, and MAF would also be affected by mixing state, especially mixing state of black carbon, while other hydrophobic components also matter (Tao et al., 2024). Authors should examine relationships between σ and χ, as well as MAF and χ. Those analysis and discussions would help and reflects impacts of mixing state on CCN activity, may be a relationship between σ and χ could be revealed.

Re: Thank you for your suggestion. Yes, it is true that using the MAF (Maximum Activation Fraction) and heterogeneity parameter σ fitted from size resolved- AR might better illustrate the effect of mixing states on CCN activity, but unfortunately, observations at IAP Summer and MHD sites lack simultaneous measurements of size resolved- AR. So here we focused on discussing the impacts of the mixing state on the critical activation diameter. The critical diameter reflects the cloud forming potential of aerosol particles at a certain supersaturation level. And as responded earlier, the mixing state index used in this study was calculated based on the probability distribution function (PDF) of κ from the in-situ measurement of the HTDMA for five diameters, which was refereed from Yuan et al. (2023). This metrics also can provide a reasonable description of the mixing state of aerosols.

We specifically discussed the relationship between hygroscopicity distribution and mixing state in **Section 3.2**, as shown in **Fig. R5 and Fig. S6**. Detail seen the above response.

[Figure]

**Fig. R5 and Fig. S6** Mixing state (χ) as a function of number fraction of the nearly hydrophobic mode (a) and more hygroscopic mode (a1) in IAP-winter; (b) and (b1) in IAP-summer; (c) and (c1) in MHD-winter; (d) and (d1) in MHD-summer.

o **Introduction** – Coastal aerosols cannot be assumed to represent marine aerosols. The introduction could follow this logic: measurements in coastal regions can provide insights into marine aerosol properties, which differ markedly from continental aerosols and have distinct climate impacts.

Re: The sentences have been revised as or see **Lines 84-102**:

"…The continental aerosols influence regional cloud formation, while coastal aerosols may provide insights into the characteristics of marine aerosols in region. The properties of marine aerosols are significantly different from those of continental aerosols, and therefore have distinct climate feedback mechanisms (Bellouin et al., 2020; Xu et al., 2024; Liu et al., 2024). However, the current models lack regional-specific mixing state parameters and usually assume uniform mixing in both environments. This could lead to large uncertainties in predicting CCN concentrations, highlighting the need for site-specific observations. For example, Ren et al. (2018) found that the impact of aerosol mixing state on CCN activation characteristics ranged from -34% to +16 % in urban atmosphere. Comparison between a fully internal mixture assumption and using the mixing state index from the particle-resolved model, Ching et al. (2017) found the obvious overestimation in CCN concentration estimation. Especially in the regions eg., Amazon Basin, Central Africa and Indonesia, the particles appeared to be more external, errors in CCN concentration would increase up to 100% (Hughes et al., 2018). A detailed exploration of mixing state on CCN concentration in global scale was conducted by Zheng et al. (2021a), and the results showed that the mixing state varied spatially with more externally mixed over the North Atlantic Ocean, off the coasts of Southern Africa, and Australia. Thus, assuming particles with internally-mixed would introduce errors in CCN concentration of 50-100% ..."

o **κ-grouping method** – The method of Yuan et al. (2023) is generally valid, but the grouping of κ=0.01 and κ=0.6 might still be improved. κ=0.01 corresponds to non-hygroscopic species such as external BC and nearly hydrophobic organic aerosols (mostly primary OA). The more hygroscopic group should contain mostly secondary organic and inorganic aerosols. At RH of 90%, κ of ammonium sulfate and nitrate is ~0.5, while SOA κ is clearly lower. Could the authors test κ_H settings for continental aerosols, as was done for marine aerosols? If κ=0.6 remains the choice, please justify.

[Figure]

**Fig. R6 or S1** Mean value of the κ–PDF for aerosols of five diameters during winter and summer periods at IAP (a and b) and MHD (c and d) sites.

Re: Thanks for your suggestion, here the heterogeneity in aerosol hygroscopicity is calculated based on the measurement of $\kappa$-PDF, differing from previous reported $\chi$ based on the chemical diversity (Ching et al., 2017; Zheng et al., 2021) by grouping two surrogate species (one with BC and POA, the other with inorganic and secondary organic aerosol species). By referred from Yuan et al. 2021, the key assumption is that the aerosol containing aerosol particles is a binary system consisting of the non- and/or less hygroscopic ($\kappa_N$ of <0.05) and more hygroscopic components ($\kappa_H$ of 0.5-0.6, referred inorganics), which corresponds to the minimum and maximum hygroscopic parameters. In ambient atmosphere, each aerosol particle in the population contains one or two of the components. As shown in Figure R6 or S1, $\kappa$-PDF at IAP can be considered the normalized aerosol number fractions varied with $\kappa$ between 0 and 0.6. and at MHD atmosphere, $\kappa$ varied between 0 and 0.8. Thus, considering of the variation in $\kappa_H$, calculation assuming $\kappa_H$ of 0.6 and 0.8 given in Figure S2 and the results show that the mixing state index do not differ significantly from those calculated assuming $\kappa_H$ of 0.6. So, the calculation assumes $\kappa_H$ of 0.6 was chosen in our study. The sensitivity of the hygroscopic parameter for the group of the hygroscopic species on the mixing state index $\chi$ was done both for the continental and coastal aerosols as seen in the revised text. See follows and **Lines 193-204**:

"…To characterize the heterogeneous distribution of the hygroscopic and non-hygroscopic components in populations (Chen et al., 2022b), we calculated the mixing state index ($\chi$) using the $\kappa$-PDF, following the methodology of Yuan et al. (2023). Two surrogate groups in a population of $N$ aerosol particles were assumed (Zheng et al., 2021a). One surrogate group consists the non- and/or slightly hygroscopic species with $\kappa_N$ of <0.05 and another group contains the more hygroscopic species with $\kappa_H$ of 0.5-0.6 (Yuan et al., 2023, referred inorganics). Ambient particles typically contain one or two of the components and the $\kappa$ lies between 0 and 0.6 at IAP or 0.8 at MHD as shown in Figure S1. Taking into account the enhanced hydrophilicity of marine aerosols at MHD site, calculation assuming $\kappa_H$ values of 0.7 and 0.8 were shown in Fig. S2. While these variations in $\kappa_H$ introduced a mean uncertainty of 8% in $\chi$ values, it did not significantly affect the seasonal or site comparisons …"

**Specific comments:**

L31 make it clear, the mixing state index here is calculated based on what kind of measurements

Re: The sentence has been revised as follows or see **Lines 28-31**:

"…This study systematically investigates the contrasting relationships between mixing states and CCN activity by combining field measurements of probability distribution function of the hygroscopicity with the algorithm of entropy at two inland and coastal sites…"

L38-40, under what levels of supersaturations (concrete value), the supersaturations typical of clouds differ for different cloud types, even for certain kind of cloud, the supersaturation vary in a wide range depending on many factors.

Re: It has been revised **Lines 44-46** and follows "… Our further quantitative analysis reveals a 0.1 increase in $\chi$ enhanced winter CCN concentrations ($N_{CCN}$) by 39–65% at the supersaturation of 0.2% ..."

L55, determine particle hygroscopicity distribution

Re: The sentence has been revised as follows or see **Lines 61-63**: "…This is problematic because mixing states directly determine particle hygroscopicity distribution and CCN estimates (Wang et al., 2010; Tao et al., 2024) ..."

L57 This statement is vague and misleading for researcher who does not understand quite well about aerosol activation. Rely on what parameters of inorganic components, size or hygroscopicity? Hygroscopicity of inorganic components of inorganic aerosols is close for ammonium nitrate and sulfate and fixed. Therefore, for internal case, still rely on aerosol size and hygroscopicity, with the hygroscopicity determined by organic aerosol hygroscopicity and organic aerosol fraction.

Therefore, for internal-mixed aerosols, the CCN activity rely on overall aerosol size, organic aerosol hygroscopicity and organic aerosol mass fraction.

For external-mixed aerosols, the CCN activity (activation fraction) also rely on overall aerosol size, organic aerosol hygroscopicity and organic aerosol mass fraction which determines the number fraction of organic aerosols.

Indeed, CCN activity depends highly on mixing state. However, for this case, CCN activity of both external- and internal-mixed aerosols depend on the organic aerosol hygroscopicity and organic aerosol mass fraction but in a different way.

Therefore, try to state in a clear way.

Re: Thanks for your suggestion, it has been revised as or See **Lines 59-68**:

"…Current models often oversimplify mixing states by assuming pure internal or external mixing (Winkler, 1973; Stevens et al., 2019; Riemer et al., 2019; Zheng et al., 2021b). This is problematic because mixing states directly determine particle hygroscopicity distribution and CCN estimates (Wang et al., 2010; Tao et al., 2024). For example, internal-mixed aerosol particles have unimodal hygroscopicity distribution, while the external-mixed particles are characterized by the bimodal/trimodal or partly overlapping structures (Spitieri et al., 2023; Liu et al., 2025). Such simplifications can lead to significant errors, e.g., Sotiropoulou et al. (2007) found that mixing state assumptions caused two-fold $N_{CCN}$ estimation errors in global models…"

L66-69, reactions during night does not significantly impact on aging thus CCN activity of aerosols? for example, heterogenous nitrate formation or multiphase sulfate formation? Why authors directly refer to photochemical reactions

Re: Thanks a lot, the statement has been revised as follows and See **Lines 74-76**:

"…Particles here undergo progressive internal mixing via photochemical process and heterogenous reactions, altering their hygroscopic properties (Ervens et al., 2010; Tao et al., 2021) …"

L71, Is this conclusion universally valid, if not, please revise as "might create unique mixing state",

Re: It has been revised as **Lines 79-80** "…Seasonal shifts in air mass sources (e.g., marine vs. continental dominance) might create unique mixing state patterns (Xu et al., 2020, 2021a) ..."

L72-74, again, night time reactions in coastal regions is not important?

Re: Revised as follows or see **Lines 80-83**: "…For instance, summer photochemical

aging and heterogenous processes in coastal areas can enhance the degree of internal mixing, while winter often retains more external mixing due to the presence of the sea-salt particles with less-hygroscopic organic matter…"

L75-76, References here are not appropriate. References here demonstrate continental aerosols and marine aerosols (mostly marine coarse aerosols) have distinct climate impacts. Marine aerosols could not be represented by coastal aerosols. As authors stated, coastal aerosols are influenced by both continental and marine air mass.

Re: The corresponding statement and references have been modified as follows or See **Lines 84-102**:

"…The continental aerosols influence regional cloud formation, while coastal aerosols may provide insights into the characteristics of marine aerosols in region. The properties of marine aerosols are significantly different from those of continental aerosols, and therefore have distinct climate feedback mechanisms (Bellouin et al., 2020; Xu et al., 2024; Liu et al., 2024). However, the current models lack regional-specific mixing state parameters and usually assume uniform mixing in both environments. This could lead to large uncertainties in predicting CCN concentrations, highlighting the need for site-specific observations. For example, Ren et al. (2018) found that the impact of aerosol mixing state on CCN activation characteristics ranged from -34% to +16 % in urban atmosphere. Comparison between a fully internal mixture assumption and using the mixing state index from the particle-resolved model, Ching et al. (2017) found the obvious overestimation in CCN concentration estimation. Especially in the regions eg., Amazon Basin, Central Africa and Indonesia, the particles appeared to be more external, errors in CCN concentration would increase up to 100% (Hughes et al., 2018). A detailed exploration of mixing state on CCN concentration in global scale was conducted by Zheng et al. (2021a), and the results showed that the mixing state varied spatially with more externally mixed over the North Atlantic Ocean, off the coasts of Southern Africa, and Australia. Thus, assuming particles with internally-mixed would introduce errors in CCN concentration of 50-100%…"

L77, both continental and coastal aerosols could only impact on regional cloud formation. Impacts of all types of aerosols on cloud formation through serve as CCN is regional.

Re: Revised as mentioned above.

L77-79, statement here should be weakened and might be misleading, aerosols at coastal regions only impact clouds near coast both on costal continental clouds and marine clouds. It could not impact on marine clouds in the vast ocean.

Re: Revised as mentioned above.

L84, Zhao et al. (2021) is an important and pioneer paper in China about this issue and should be cited, the history of the entropy philosophy should also be included, for example, Riemer and West (2013).

Re: Thank you, the definition of "mixing state" has been added in the revised text and see **Lines 103-120:**

"…Therefore, for quantifying the aerosol mixing state in the ambient atmosphere, we apply the algorithm of entropy proposed by Riemer and West (2013) to investigate the aerosol heterogeneity. This index has been applied to quantify the mixing state more

reasonably both in field campaigns (Zhao et al., 2021; Yuan et al., 2023) and model simulations (Ching et al., 2016; Zheng et al., 2021a). However, most studies focused on quantifying the particle heterogeneity in composition (Ching et al., 2019; Fierce et al., 2020; Zhao et al., 2021). Here we concentrated on evaluating the heterogeneity in aerosol hygroscopicity for sub-micron particles, which directly related to CCN budget. By refereeing to Yuan et al. 2023, the heterogeneity in hygroscopicity was investigated by combining in-situ measurements of probability distribution function of the hygroscopicity with the algorithm of entropy. Briefly, the mixing state index $\chi$, is devised based on the concept of information entropy concerning the distribution of hygroscopicity across the aerosol population. It varies between 0 (external mixing completely) and 1 (internal mixing completely). By integrating inland and coastal measurements, this study will focus on addressing two key gaps, (1) How continental vs. marine-dominated environments shape aerosol mixing states and CCN activity; (2) Whether $\chi$-based CCN parameterizations show regional dependencies, providing critical constraints for climate models …"

L144-147, CCN measurements conducted at what levels of supersaturations.

Re: Revised and See **Lines 184-191**:

"…The CCN number concentrations were quantified at both sites using a Droplet Measurement Technologies CCN counter (DMT-CCNc) (Lance et al., 2006). The instrument's supersaturation (SS) settings were carefully calibrated before and after each campaign using ammonium sulfate aerosol following Rose et al. (2008). Four effective supersaturations (SS) were 0.14%, 0.23%, 0.40% and 0.76% at IAP site. Four SS levels were 0.25%, 0.5%, 0.75% and 1% at MHD site with an uncertainty of ±0.03%. Using measurements at set supersaturation of 0.2% as an example explores the CCN activity in the following discussions…"

L226-227, this speculation was based on what evidence? Please include more discussions.

Re: The sentence has been revised as and See **Lines 271-274**: "…For example, $\chi$ for 40 nm particles decreases as PM increases at IAP site (Fig. 3c). The elevated particle heterogeneity mainly arises from the locally primary emissions, corresponding to the enhanced primary organic emissions as shown in Fig. S4. It appeared more pronounced during evening rush hours …"

[Figure]

**Fig. R7 or S4.** Diurnal variation of the particle number size distribution and CN number concentration during winter (a) and summer periods at IAP site (a1), particle matter concentration and mass fraction of the chemical components (b and b1), the mixing state index (χ) at 40 and 150 nm (c and c1), number fraction of nearly hydrophobic mode (NH) (d and d1) and more hygroscopic mode (MH) particles (e and e1).

L248-251, The limited measurements could suggest conclusions across seasons? Please explore more the relationships between diurnal variations of accumulation mode aerosol hygroscopicity distribution and aerosol chemical composition evolutions and put it into the context of existing literatures.

Re: Thanks a lot, it has been revised as follows and See **Lines 295-301**: "…Conversely, the χ for accumulation-mode particles showed minimal diurnal variations both in IAP-winter and IAP-summer. This is mainly due to the dominant hygroscopic mode for 150 nm particles (Fig. 4g), especially during summer, which is mainly from secondary formation or aging of the primary particles (such as the transformation from primary organic aerosol (POA) to secondary organic aerosol (SOA) in Fig. S4) (Wang et al., 2019; Fan et al., 2020) …"

L249, I did not see pronounced diurnal cycles especially for Mace Head measurements, how to defined pronounced? Please use objective statement, for example "diurnal

variations of xx is shown in Fig.x" ???

Re: Revised as follows and See **Lines 286-287**: "…Diurnal variations of mixing state metrics (Dα, Dγ, Gf-PDF and χ) at IAP and MHD sites are shown in Figure 4…"

L265, the "spread factor" is what? Please make it clear

Re: Revised and See **Lines 315-317**: "…This trend demonstrates a strong alignment with the spread factor (used as a measure of particle mixing state) documented by Xu et al. (2021a) …"

L267-268, biogenic origin is inferred from what clue? At least add the reference

Re: Revised and See **Lines 319-325**: "…In winter, the Gf-PDF diurnal profiles of both Aitken and accumulation mode particles showed bimodal distribution (Fig. 4e2-g2) as evident by the number fraction of nearly-hydrophobic and more hygroscopic modes (Fig. S5). The NH mode was likely to be the anthropogenic organic matter and biogenic origin from marine mass (Xu et al., 2020), especially for the Aitken mode. The more hygroscopic and sea salt mode was mostly contributed from the nss-sulfate and sea salt in winter (Xu et al., 2021a) …"

L276 to 278, this speculation is not convincing at all. The unimodal distribution peaked at gf of ~1.7, meaning that the kappa peaked at ~0.5, the secondary formation of organic aerosol as demonstrated by the cited reference Jimenez et al. would substantially reduce the aerosol hygroscopicity, which is contrast with the average distribution here. The results shown in Fig.3d also demonstrate that the mixing state index would decrease as PM increase with increase organic aerosol fraction, authors should plot the average Kappa distribution under different PM levels at the Mace Head, and authors might find that the unimodal distribution prevail only for small PM conditions in summer.

I guess that the unimodal distribution in summer is due to the marine air mass not because of the aging, the aging in summer would undermine the unimodal distribution.

[Figure]

**Fig. R8** Average κ-PDF for 35 nm and 165 nm under different PM levels (0-2, 2-4, 4-6, 6-8 μg/m³) in MHD-Summer.

[Figure]

**Fig. R9 or S5** Diurnal variation of the particle number size distribution and CN number concentration during winter (a) and summer periods at MHD site (a1), particle matter concentration and mass fraction of the chemical components (b and b1), the mixing state index ($\chi$) at 35 and 165 nm (c and c1), number fraction of nearly hydrophobic mode (NH) (d and d1), more hygroscopic mode (MH) (e and e1) and sea salt mode (SS) particles (f and f1).

Re: Thank you for your suggestion. The average hygroscopicity distribution in summer under different PM levels was shown in the **Fig. R9**. It can be observed that with the increase of PM, $\kappa$-PDF for Aitken-mode particles exhibit a bimodal distribution, mainly due to the disturbance of anthropogenic organic matter, while the accumulation mode shows a unimodal distribution, indicating the promoting effect of aging. And from **Fig. R9**, in MHD-summer, the $\chi$ exhibited a clear increase at midday again in line with the assumption of increased photochemical activity by turning NH particles into MH ones. Overall, the higher hygroscopicity and more internal mixed in MHD-summer were

mainly associated with increased contribution of nss-$SO_4^{2-}$, the decrease of the organic matter, and the promotion of photo-oxidation activity and aging process (Xu et al., 2021).

Revised see follows and **Lines 330-338:** "…In contrast, summer observations revealed that Gf-PDFs of both Aitken and accumulation mode particles transitioned to unimodal distributions, signifying particles in summer had more homogeneous composition with a large extent of internal mixing particles (with higher χ). Such diurnal trend in Gf-PDFs was consistent along with the high number fraction of MH-mode and low NH-mode (Fig. S5). The higher hygroscopicity and MH mode in summer were largely driven by the enhancement of sulfate and decrease of organic matter (Fig. S5). And a clear shift from NH to MH mode at midday might further demonstrate the promotion of photochemical aging in summer (Xu et al., 2021a) ..."

L286, what kind of aerosol properties

Re: Revised. See follows and **Lines 415-418:** "…The variations of particle size and chemical composition with the increments of χ (ranging from 0 to 1 with the step of 0.1) were illustrated in Figure 7, presenting key insights of two fundamental determinants of CCN activity (Dusek et al., 2006) …"

L289, reference (Ren et al., 2018) is not appropriate, reference such as Dusek et al. (2006) serves better

Re: Revised.

L291 Figure captions of Figure 5 is not clear,

Re: Figure 5 has been adjusted as Figure 7 in the revised text, and the caption has been revised see follows: "Figure 7. Comparison of the average particle number size distribution (PNSD) in different mixing state index (χ) (a), CN number concentration ($N_{CN}$) as a function of χ (b), Critical diameter ($D_{cri}$) at S=0.2% and mass fraction of chemical composition as a function of χ (c), CCN number concentration ($N_{CCN}$) (d) and activation ratio (AR) at S=0.2% a function of χ (e)."

L294-296, speculation, how could you attribute to new particle formation? I did see the NPD characteristics embedded in the PNSD of IAP summer

Re: Revised see follows and **Lines 429-433:** "…Notably, new particle formation events frequently occurred in IAP-summer (Fig. S8), corresponding the gradually increase of χ. And the χ for Aitken-mode is significantly larger than the accumulation-mode particles during this period. Thus, $N_{CN}$ exhibits a sustained slight increase as the degree of the internal mixing increases in IAP-summer ..."

[Figure]

**Fig. R10 or S8** Time series of the particle number size distribution (a), χ for 40 and 150 nm (b), particle matter mass concentration and the difference of χ between 150 and 40 nm (c) in IAP-summer.

L298, only mass fraction of water-soluble components? Depend on aerosol hygroscopicity which is determined by hygroscopicity and mass fractions of water-soluble components.

Re: It has been revised as and see **Lines 434-437**: "…The critical diameter ($D_{cri}$)—defined as the minimum size for activation at a given supersaturation—depends on aerosol hygroscopicity. This hygroscopicity is determined by both the hygroscopicity and the mass fraction of soluble components (Petters and Kreidenweis, 2007) …"

L299, typical of what types of cloud? revise "Using a typical cloud supersaturation of 0.2% as a case study" as "Using measurements at supersaturation of 0.2% as an example".

Re: Thanks for your suggestion, the sentence has been revised.

L300 "decreases with increasing highly hygroscopic inorganic aerosol components (e.g., sulfate, nitrate)", the key point is inorganic aerosol components with high hygroscopicity, not water soluble. Many water-soluble substances have small hygroscopicity due to high molecular weight and small van't Hoff factor. By the way, water soluble might correspond to very small hygroscopicity (Chen et al., 2019).

Re: The sentence has been revised as follows or **Lines 437-439**"…Using measurements at supersaturation of 0.2% as an example, Fig. 7c shows that $D_{cri}$ decreases with increasing highly hygroscopic inorganic components (e.g., sulfate, nitrate) in the inland atmosphere …"

L315, the dominant role of aerosol size on CCN activity should cite the paper of Dusek et al. (2006).

Re: Revised.

Chen, J., Lee, W.-C., Itoh, M., and Kuwata, M.: A Significant Portion of Water-Soluble

Organic Matter in Fresh Biomass Burning Particles Does Not Contribute to Hygroscopic Growth: An Application of Polarity Segregation by 1-Octanol–Water Partitioning Method, Environmental science & technology, 53, 10034-10042, 10.1021/acs.est.9b01696, 2019.

Dusek, U., Frank, G. P., Hildebrandt, L., Curtius, J., Schneider, J., Walter, S., Chand, D., Drewnick, F., Hings, S., Jung, D., Borrmann, S., and Andreae, M. O.: Size Matters More Than Chemistry for Cloud-Nucleating Ability of Aerosol Particles, Science, 312, 1375-1378, 10.1126/science.1125261, 2006.

Riemer, N., and West, M.: Quantifying aerosol mixing state with entropy and diversity measures, Atmos. Chem. Phys., 13, 11423-11439, 10.5194/acp-13-11423-2013, 2013.

Rose, D., Gunthe, S. S., Mikhailov, E., Frank, G. P., Dusek, U., Andreae, M. O., and Pöschl, U.: Calibration and measurement uncertainties of a continuous-flow cloud condensation nuclei counter (DMT-CCNC): CCN activation of ammonium sulfate and sodium chloride aerosol particles in theory and experiment, Atmos. Chem. Phys., 8, 1153-1179, 10.5194/acp-8-1153-2008, 2008.

Tao, J., Luo, B., Xu, W., Zhao, G., Xu, H., Xue, B., Zhai, M., Xu, W., Zhao, H., Ren, S., Zhou, G., Liu, L., Kuang, Y., and Sun, Y.: Markedly different impacts of primary emissions and secondary aerosol formation on aerosol mixing states revealed by simultaneous measurements of CCNC, H(/V)TDMA, and SP2, Atmos. Chem. Phys., 24, 9131-9154, 10.5194/acp-24-9131-2024, 2024.

Zhao, G., Tan, T., Zhu, Y., Hu, M., and Zhao, C.: Method to quantify black carbon aerosol light absorption enhancement with a mixing state index, Atmos. Chem. Phys., 21, 18055-18063, 10.5194/acp-21-18055-2021, 2021.

**References for the Response**

Riemer, N., Ault, A. P., West, M., Craig, R. L., Curtis, J. H.: Aerosol mixing state: Measurements, modeling, and impacts, Reviews of Geophysics., 57(2), 187-249, https://doi.org/10.1029/2018RG000615, 2019.

Bhattu, D., Tripathi, S. N.: CCN closure study: Effects of aerosol chemical composition and mixing state, Journal of Geophysical Research: Atmospheres., 120(2), 766-783, https://doi.org/10.1002/2014JD021978, 2015.

Ching, J., Zaveri, R. A., Easter, R. C., Riemer, N., Fast, J. D.: A three-dimensional sectional representation of aerosol mixing state for simulating optical properties and cloud condensation nuclei, Journal of Geophysical Research: Atmospheres., 121(10), 5912-5929, https://doi.org/10.1002/2015JD024323, 2016.

Ching, J., Fast, J., West, M., Riemer, N.: Metrics to quantify the importance of mixing state for CCN activity, Atmospheric Chemistry and Physics., 17(12), 7445-7458, https://doi.org/10.5194/acp-17-7445-2017, 2017.

Ching, J., Adachi, K., Zaizen, Y., Igarashi, Y., Kajino, M.: Aerosol mixing state revealed by transmission electron microscopy pertaining to cloud formation and human airway deposition, npj Climate and Atmospheric Science., 2(1), 22, https://doi.org/ 10.1038/s41612-019-0081-9, 2019.

Collins, D. B., Ault, A. P., Moffet, R. C., Ruppel, M. J., Cuadra-Rodriguez, L. A., Guasco, T. L., Corrigan, C. E., Pedler, B. E., Azam, F., Aluwihare, L. I., Bertram, T. H., Roberts, G. C., Grassian, V. H., Prather, K. A.: Impact of marine biogeochemistry on the chemical mixing state and cloud forming ability of nascent sea spray aerosol, Journal of Geophysical Research: Atmospheres.,

118(15), 8553-8565, https://doi.org/10.1002/jgrd.50598, 2013.

Ervens, B., Cubison, M. J., Andrews, E., Feingold, G., Ogren, J. A., Jimenez, J. L., Quinn, P. K., Bates, T. S., Wang, J., Zhang, Q., Coe, H., Flynn, M., Allan, J. D.: CCN predictions using simplified assumptions of organic aerosol composition and mixing state: a synthesis from six different locations, Atmospheric Chemistry and Physics., 10(10), 4795-4807, https://doi.org/10.5194/acp-10-4795-2010, 2010.

Petters, M. D., Kreidenweis, S. M.: A single parameter representation of hygroscopic growth and cloud condensation nucleus activity, Atmospheric Chemistry and Physics., 7(8), 1961-1971, https://doi.org/10.5194/acp-7-1961-2007, 2007.

Riemer, N., Ault, A. P., West, M., Craig, R. L., Curtis, J. H.: Aerosol mixing state: Measurements, modeling, and impacts, Reviews of Geophysics., 57(2), 187-249, https://doi.org/10.1029/2018RG000615, 2019.

Riemer, N. and West, M.: Quantifying aerosol mixing state with entropy and diversity measures, Atmos. Chem. Phys., 13, 11423–11439, https://doi.org/10.5194/acp-13-11423-2013, 2013.

Ren, J., Zhang, F., Wang, Y., Collins, D., Fan, X., Jin, X., Xu, W., Sun, Y., Cribb, M., Li, Z.: Using different assumptions of aerosol mixing state and chemical composition to predict CCN concentrations based on field measurements in urban Beijing, Atmospheric Chemistry and Physics., 18(9), 6907-6921, https://doi.org/10.5194/acp-18-6907-2018 2018.

Stevens, R., and Dastoor, A.: A Review of the Representation of Aerosol Mixing State in Atmospheric Models, Atmosphere., 10, 168, https://doi.org/10.3390/atmos10040168, 2019.

Shi, Z., Vu, T., Kotthaus, S., Harrison, R. M., Grimmond, S., Yue, S., Zhu, T., Lee, J., Han, Y., Demuzere, M., Dunmore, R. E., Ren, L., Liu, D., Wang, Y., Wild, O., Allan, J., Acton, W. J., Barlow, J., Barratt, B., Beddows, D., Bloss, W. J., Calzolai, G., Carruthers, D., Carslaw, D. C., Chan, Q., Chatzidiakou, L., Chen, Y., Crilley, L., Coe, H., Dai, T., Doherty, R., Duan, F., Fu, P., Ge, B., Ge, M., Guan, D., Hamilton, J. F., He, K., Heal, M., Heard, D., Hewitt, C. N., Hollaway, M., Hu, M., Ji, D., Jiang, X., Jones, R., Kalberer, M., Kelly, F. J., Kramer, L., Langford, B., Lin, C., Lewis, A. C., Li, J., Li, W., Liu, H., Liu, J., Loh, M., Lu, K., Lucarelli, F., Mann, G., McFiggans, G., Miller, M. R., Mills, G., Monk, P., Nemitz, E., O'Connor, F., Ouyang, B., Palmer, P. I., Percival, C., Popoola, O., Reeves, C., Rickard, A. R., Shao, L., Shi, G., Spracklen, D., Stevenson, D., Sun, Y., Sun, Z., Tao, S., Tong, S., Wang, Q., Wang, W., Wang, X., Wang, X., Wang, Z., Wei, L., Whalley, L., Wu, X., Wu, Z., Xie, P., Yang, F., Zhang, Q., Zhang, Y., Zhang, Y., and Zheng, M.: Introduction to the special issue "In-depth study of air pollution sources and processes within Beijing and its surrounding region (APHH-Beijing)", Atmos. Chem. Phys., 19, 7519–7546, https://doi.org/10.5194/acp-19-7519-2019, 2019.

Tao, J., Luo, B., Xu, W., Zhao, G., Xu, H., Xue, B., Zhai, M., Xu, W., Zhao, H., Ren, S., Zhou, G., Liu, L., Kuang, Y., and Sun, Y.: Markedly different impacts of primary emissions and secondary aerosol formation on aerosol mixing states revealed by simultaneous measurements of CCNC, H(/V)TDMA, and SP2, Atmos. Chem. Phys., 24, 9131–9154, https://doi.org/10.5194/acp-24-9131-2024, 2024.

Xu, W., Ovadnevaite, J., Fossum, K. N., Lin, C., Huang, R.-J., O'Dowd, C., Ceburnis, D.: Seasonal trends of aerosol hygroscopicity and mixing state in clean marine and polluted continental air masses over the Northeast Atlantic, Journal of Geophysical Research: Atmospheres., 126, e2020JD033851, https://doi.org/10.1029/2020JD033851, 2021.

Yuan, L., and Zhao, C.: Quantifying particle-to-particle heterogeneity in aerosol hygroscopicity,

Atmospheric Chemistry and Physics., 23, 3195-3205, https://doi.org/10.5194/acp-23-3195-2023, 2023.

Yang, F., Chen, H., Du, J., Yang, X., Gao, S., Chen, J., Geng, F.: Evolution of the mixing state of fine aerosols during haze events in Shanghai, Atmospheric Research., 104: 193-201, https://doi.org/10.1016/j.atmosres.2011.10.005, 2012.

Zheng, Z., Curtis, J. H., Yao, Y., Gasparik, J. T., Anantharaj, V. G., Zhao, L., West, M., Riemer, N.: Estimating submicron aerosol mixing state at the global scale with machine learning and Earth system modeling, Earth and Space Science., 8(2), e2020EA001500, https://doi.org/10.1029/2020EA001500, 2021.

Zhao, G., Tan, T., Zhu, Y., Hu, M., and Zhao, C.: Method to quantify black carbon aerosol light absorption enhancement with a mixing state index, Atmospheric Chemistry and Physics., 21, 18055–18063, https://doi.org/10.5194/acp-21-18055-2021, 2021.